# Metabolite changes during developmental transitions in *Adonis amurensis* Regel et Radde flowers: Insights from HPLC-MS analysis

**Zun Xingzun[1], Wang Hongtao[2]\***

1 College of Landscape Architecture, Changchun University, Chaoyang District, Changchun City, Jilin, China, 2 Changbiashan Key Laboratory of Biological Germplasm Resources Evaluation and Application, Tonghua Normal University, Dongchang District, Tonghua City, Jilin, China

\* hongtaowang@thnu.edu.cn

**Data Availability Statement:** All relevant data are within the manuscript and its Supporting Information files.

## Abstract

*Adonis amurensis* Regel et Radde is a remarkable and important spring ephemeral plant and gained considerable attention because of its remarkable medicinal properties. Extensive research has been conducted on its therapeutic applications, physical characteristics, flowering patterns, reproductive, cultural and molecular biology. However, there is a lack of comprehensive understanding regarding the metabolic changes associated with flower developmental stages. This study was designed to investigate the changes in metabolites and their interrelationships at five distinct developmental stages of *A. amurensis* flower: Flower Primordium (FP), Sepal Stage (SE), Perianth Primordium (PE), Stamens Stage (SE), and Pistil Stage (PI). High-performance liquid chromatography coupled with mass spectrometry (HPLC-MS) was utilized to investigate and characterize the metabolites associated with specific flower developmental stages. The various stages of flower development exerted a substantial influence on both the quantity and composition of metabolites present, signifying significant changes in the types and quantities of metabolites throughout the developmental progression of the flower. Metabolite Set Enrichment Analysis (MSEA) and annotation via the KEGG database highlighted enriched pathways such as flavonoid biosynthesis and plant hormone signal transduction, which are crucial for flower maturation. The highest number of differentially expressed metabolites was identified between the SE and PI stages, emphasizing a marked appreciation in metabolite expression linked to the development of reproductive organs. Key pathways such as flavonoid biosynthesis and plant hormone signal transduction were markedly enriched, underscoring their roles in flower maturation and potential pharmacological applications. Our research not only helps us in understanding the metabolomic dynamics during the flower development of *A. amurensis* but also emphasizes the potential pharmacological implication of stage-specific metabolites. Identifying these metabolites can help targeted bioprospecting and optimization of extraction methods to tackle the plant's full therapeutic potential, particularly in the development of treatments for cardiac insufficiency, edema, and possibly cancer.

**Funding:** This research was supported by "Growing Up Scholar Climbing Program, Changchun University" No. zkp202205. The funding organizations had no role in study design or concept; collection, management, analysis, or interpretation of data; or preparation, review, or approval of manuscript. There was no additional funding resources for this study.

**Competing interests:** No competing interests.

## Introduction

*Adonis* belongs to the Ranunculaceae family and comprises both annual and perennial plants. It is a genus that encompasses a wide distribution, with approximately 26 to 30 species found in Northeast China, North Korea, and the far east of Russia [1–3]. Among the various species within the Adonis genus, *Adonis amurensis* Regel et Radde holds significant importance as a spring ephemeral plant. This specific species has been widely employed in medicinal practices, either in its entirety as a whole plant or through extracts derived from its various parts. Its primary application lies in the treatment of cardiac insufficiency and edema [4]. Furthermore, research has revealed that extracts from *A. amurensis* possess notable properties, including of inhibiting cell growth in human umbilical vein endothelial cells, suppression of neovascularization, and potential anti-cancer effects [5, 6]. Because of its bright color and high ornamental value, it has good application prospects in landscape design [7]. Previous studies on *A. amurensis* have primarily centered around its morphology, flowering characteristics, reproductive and cultural biology, and molecular biology, among other relevant aspects [8]. Notably, *A. amurensis* flower buds require prolonged low temperatures for growth [9] and are affected by the concentration of gibberellins and cytokinins [5]. The Karyotype and genome size of *A. amurensis* has recently been studied [10].

Flowers are the most complicated structures found in plants as they encompass the reproductive organs responsible for producing male (stamens) and female (ovaries) reproductive bodies. The stamens produce anthers containing pollen, while the ovaries contain ovules, which develop into seeds upon fertilization [11]. The process of flowering stands as a pivotal developmental stage in the life cycle of flowering plants, representing the shift from vegetative growth to reproductive development. It is a vital process for successfully producing seeds in flowering plants [12].

While the identification and functional characterization of flowering genes in model plants have provided valuable insights [13, 14], extending these studies to plants that flourish in unique environments or under extreme conditions is crucial. Such investigations are necessary to deepen our understanding of the molecular mechanisms underlying flowering regulation. Flower development is a primarily important process in the life cycle of angiosperms, marking the transition from vegetative growth to reproductive development. *A. amurensis*, a perennial herbaceous flower belonging to the Ranunculaceae family, presents an intriguing case study. In northeast China, *A. amurensis* is known to bloom before the ice and snow melt during the spring, even when temperatures drop as low as -15˚C [15]. Several other species have been studied under harsh environmental conditions [16]. Despite this remarkable adaptation, the genetic regulatory mechanisms governing flower developmental stages in *A. amurensis* under extremely low temperatures remain unclear. Previously, we and others have identified the special characteristics of the flowering developmental stages of this species, further research is needed to unravel the complex genetic pathways involved in flowering regulation at each developmental stage in *A. amurensis* under these challenging environmental conditions [15, 17]. Transcriptomics and proteomics have been extensively applied to study the physiological aspects at the molecular level to identify individual factors involved [18, 19]. Metabolomics is one similar technology used to investigate metabolomes that play an important role in physiological processes. Metabolomics involves the comprehensive analysis of small molecules, or metabolites, present in a biological system [20, 21]. By examining the changes in metabolite profiles during different flowering stages, metabolomics provides insights into the metabolic pathways and regulatory networks underlying this developmental process [22]. It helps identify critical metabolites associated with flowering and can uncover novel genes and enzymes involved in flower development. Integrating metabolomic data with transcriptomic and

proteomic information offers a holistic understanding of the molecular mechanisms governing flowering in plants [23]. Despite extensive research is conducted on the flowering of *A. amurensis*, there remains a gap in understanding the metabolic changes associated with flower development. Understanding the metabolic pathways and regulatory networks that enable such unique flowering characteristics under harsh conditions can provide valuable insights into the genetic mechanisms of flower development. In this study, high-performance liquid chromatography coupled with mass spectrometry (HPLC-MS) was used to dig out the metabolites associated with specific flower developmental stages of *A. amurensis*, such as Flower Primordium (FP), Sepal Stage (SE), Perianth Primordium, (PE) Stamens Stage (SE) and Pistil Stage (PI). The data obtained were analyzed using Metabolite Set Enrichment Analysis (MSEA) and annotated using the KEGG (Kyoto Encyclopedia of Genes and Genomes) database. Our findings are valuable to enhance our understanding of the complex metabolic dynamics that facilitate flower development under extreme environmental conditions, governing flower development and the transitions between various developmental stages. We aim to uncover the offering a holistic understanding of the molecular mechanisms governing this process. It may be helpful in uncovering the potential pharmacological implications of stage-specific metabolites.

## Materials and methods

### Plant materials

This research was conducted at the Tonghua Normal University (TNU) field station in the Tonghua section of the Southeast Changbai Mountains. The field station is located at 41° 44' 47.04" N, 125° 58' 49.63" E, with an elevation of 429 meters above sea level. The region has a north-temperate continental monsoon climate characterized by cold, dry winters and hot, humid summers. The average annual temperature ranges from 5.5 to 6.0°C, with a frost-free period of 140 to 171 days. The average annual precipitation ranges from 893 to 1083.8 mm, with the majority falling in July and August. Flower samples of *A. amurensis* were collected at five distinct developmental stages: Flower Primordium (FP), Sepal Stage (SE), Perianth Primordium (PE), Stamens Stage (ST), and Pistil Stage (PI), with three biological replicates per stage. The identification of these distinct stages was based on our previous research, where we thoroughly characterized these developmental stages of *A. amurensis*; identified these stages as critical points in the flower's development, representing significant morphological and physiological changes [17]. The plant materials were identified by Prof. Wang Hongtao, and kept at the publicly accessible herbarium. The detailed information is as follows. Herbarium, School of Life Sciences in Tonghua Normal University, Specimen No. THUN—MGK—03 – 001. SEM images of each stage and a fully bloomed flower are provided in Fig 1. These stages were chosen to provide a comprehensive understanding of the metabolic changes associated with key developmental transitions. Samples were harvested in the early morning using sterilized scissors and placed into pre-labeled containers. Immediately after collection, the samples were flash-frozen in liquid nitrogen in the field, transported to the laboratory in liquid nitrogen containers, and stored at -80°C until further processing. In the laboratory, the frozen samples were lyophilized, grounded into a fine powder using a pre-cooled mortar and pestle and a mixture was obtained by following other researchers with minor modifications [24]. The mixture was sonicated for 30 minutes at room temperature, centrifuged at 12,000 rpm for 10 minutes at 4°C, and the supernatants were filtered through a 0.22 μm membrane filter. The filtered extracts were aliquoted and stored at -80°C until analysis by HPLC-MS. Each stage was treated as a separate group for analysis. The sample grouping and corresponding information are given in Table 1.

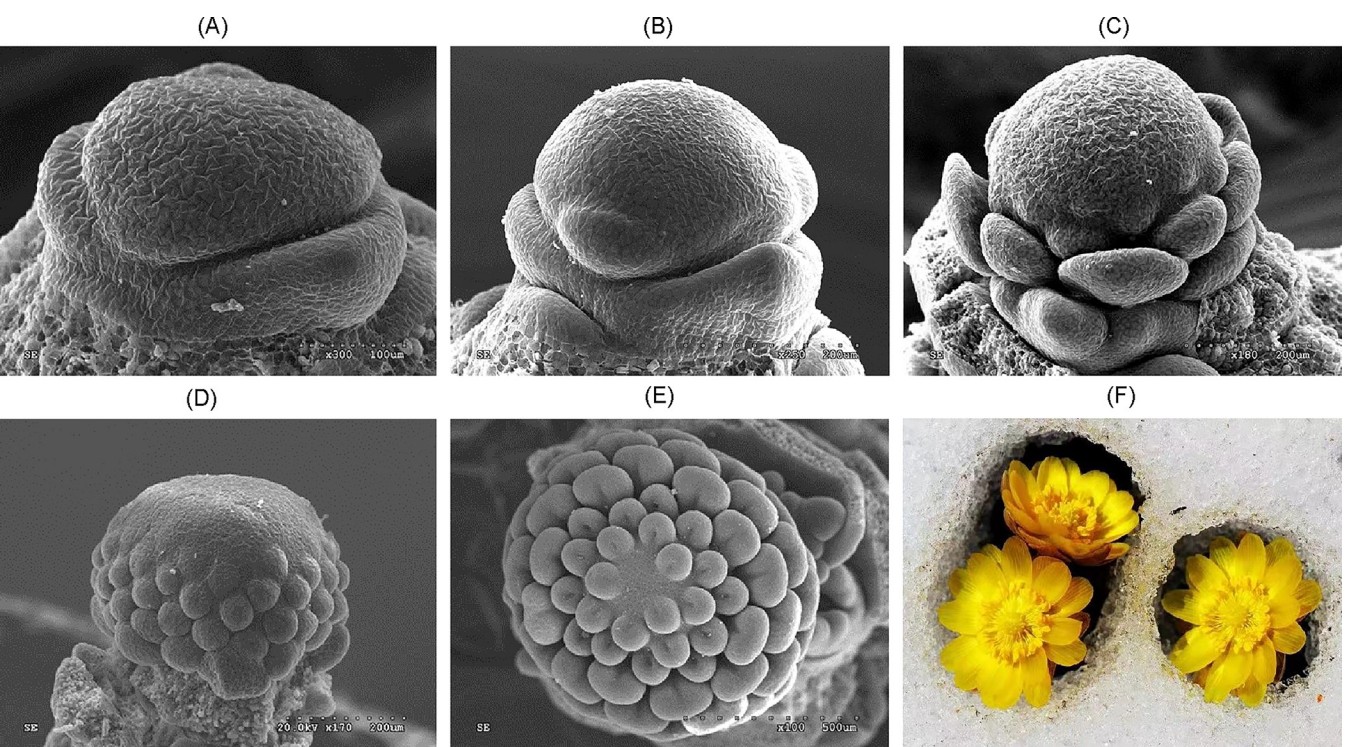

**Fig 1. Overview of *A. amurensis* flower organogenesis [17].** (A-E) Scanning electronic microscopic (SEM) images of *A. amurensis* developing flower at flower primordium (FP), sepal (SE), Perianth Primordium (PE), Stamen stage (ST), and Pistil stage (PI) differentiation stages, respectively. (F) An image of *A. amurensis* flower during early spring.

## High-performance liquid chromatography/mass spectrometry conditions

Data acquisition instrumentation systems include Ultra Performance Liquid Chromatography (UPLC) (ExionLC™ AD, https://sciex.com.cn/) and tandem Mass Spectrometry (MS/MS) (Applied Biosystems 4500) QTRAP, https://sciex.com.cn/) were used. Mass spectrometry

**Table 1. The information of the samples and groups.**

| Group No. | Flower developmental stage | Replicates | Group |
|:---:|:---:|:---:|:---:|
| 1 | Flower Primordium | FP1 | FP |
|   |   | FP2 |   |
|   |   | FP3 |   |
| 2 | Sepal Stage | SE1 | SE |
|   |   | SE2 |   |
|   |   | SE3 |   |
| 3 | Perianth Primordium | PE1 | PE |
|   |   | PE2 |   |
|   |   | PE3 |   |
| 4 | Stamens Stage | ST1 | ST |
|   |   | ST2 |   |
|   |   | ST3 |   |
| 5 | Pistil Stage | PI1 | PI |
|   |   | PI2 |   |
|   |   | PI3 |   |

conditions mainly included Electrospray ionization (ESI) temperature 550˚C; Ion spray voltage (IS) 5500 V (positive mode) / -4500 V (negative ion mode); Ion source gases I (GSI), gas II (GSII), and curtain gas (CUR) were set to 50, 60, and 25 psi, respectively, and the collision-induced ionization parameters were set to high. QQQ scan uses MRM mode and sets the collision gas (nitrogen) to medium. Through further optimization of decluttering potential (DP) and collision energy (CE), the DP and CE of each MRM ion pair were monitored.

## Qualitative and quantitative analysis of metabolites

Using the MWDB (Metware database), substances are characterized based on secondary spectral information. The MWDB is a comprehensive metabolomics database containing extensive metabolite information, which ensures high reliability and accuracy in metabolite identification and annotation and has been extensively utilized by the researchers [25, 26]. Isotopic and repeat signals containing $K^+$, $Na^+$, $NH_4^+$ are removed, as well as repeat signals of fragment ions from more significant molecular weight substances. Metabolite quantification is performed using triple quadrupole mass spectrometry in multiple reaction monitoring mode (MRM). MRM screens precursor ions and excludes other molecular-weight substances to eliminate interference. The collision chamber induces and fractures precursor ions, generating fragment ions. Triple quadrupole filtering selects characteristic fragment ions to eliminate non-target ion interference, improving accuracy and repeatability. Metabolite profiling data from different samples are integrated and corrected [27]. Quality Control (QC) samples were prepared by pooling equal aliquots from each biological replicate of the samples. This pooled sample was then processed in the same manner as the study samples, including lyophilization, grinding, sonication, centrifugation, and filtration. QC samples were inserted every 10 test samples during instrumental analysis to monitor the repeatability and stability of the analytical process. The impact of QC samples on data quality was assessed by evaluating the reproducibility of metabolite measurements. Overlapping total ion flow patterns (TIC plots) of different QC samples were used to assess the technical replicates' repeatability. The coefficient of variation (CV) values for the metabolites in the QC samples were calculated, with a lower CV indicating higher stability and reproducibility. In our study, over 85% of substances in the QC samples had CV values below 0.5, and over 75% had CV values below 0.3, indicating excellent stability and reproducibility of the data.CV value distribution plot for all samples

The CV value (coefficient of variation) reflects data dispersion, calculated as the ratio of standard deviation to mean. The ECDF (Empirical Cumulative Distribution Function) analyzes the frequency of substances with CV values below a reference value. More substances with lower CV values in QC samples indicate more excellent data stability. In QC samples, over 85% of substances have CV values below 0.5, indicating high stability. Additionally, over 75% of substances in QC samples have CV values below 0.3, indicating very stable experimental data.

## KEGG annotation and pathway enrichment analysis of metabolites

Metabolites were annotated using the KEGG compound database (http://www.kegg.jp/kegg/compound/) and mapped to the KEGG pathway database (http://www.kegg.jp/kegg/pathway.html). Significant metabolite-mapped pathways underwent metabolite sets enrichment analysis (MSEA) using the hypergeometric test for determining significance.

## Statistical analysis

Metabolite data were analyzed using Analyst 1.6.3 software. For multivariate data analysis, we employed Orthogonal Partial Least Squares Discriminant Analysis (OPLS-DA) to enhance

sample differentiation in the metabolome. OPLS-DA was chosen for its ability to maximize the separation between groups by removing variation unrelated to class separation, thereby providing clearer insights into the differences in metabolite profiles between developmental stages. The Variable Importance in Projection (VIP) parameter was used to assess the significance of each metabolite in the OPLS-DA model. Metabolites with VIP $\geq$ 1 and fold change $\geq$ 2 or $\leq$ 0.5 were considered differential for group discrimination. For univariate statistical analysis, SPSS 20.0 was used. The selection of SPSS was based on its robust capabilities for performing comprehensive statistical analyses and its widespread acceptance in scientific research. Data were presented as mean ± SD. Group differences were evaluated with analysis of variance (ANOVA), which is a powerful method for comparing means among multiple groups. P $\leq$ 0.05 was considered statistically significant.

## Results

### Differential metabolite analysis and cluster heatmap analysis

The metabolomic profiling of *A. amurensis* flowers at five distinct developmental stages— Flower Primordium (FP), Sepal Stage (SE), Perianth Primordium (PE), Stamens Stage (ST), and Pistil Stage (PI)—revealed significant changes in metabolite composition and abundance. High-performance liquid chromatography coupled with mass spectrometry (HPLC-MS) identified the differentially expressed metabolites. Statistical significance of the differences in metabolite levels between stages was assessed using ANOVA, with p-values $\leq$ 0.05 considered significant. Additionally, false discovery rates (FDR) were calculated to correct for multiple comparisons, with FDR $\leq$ 0.05 indicating statistically significant metabolites. Table 2 shows the detailed description of these differentially expressed metabolites compared to each group. We found that SE vs PI had the highest number (453) of significantly differentiated metabolites, while PE vs ST had the lowest number of differentially expressed metabolites. A heatmap cluster (Fig 2) shows the groups' relationships. Metabolite distribution is sample-specific, with varying types and proportions in each group. Metabolite composition changes across processes or biological events. Metabolite ratio analysis illustrates the major metabolite distribution in the sample, as shown in Fig 3.

### Metabolome data quality analysis via coefficient (CV) value distribution

The coefficient of variation (CV) reflects data dispersion, calculated as the ratio of standard deviation to mean. The Empirical Cumulative Distribution Function (ECDF) analyzes CV frequency of substances below a reference value. More substances with lower CV values in QC

**Table 2. Statistics of the number of differential metabolites.**

| S. No. | Group name | All significant differences | Down-regulated | Up-regulated |
|--------|-----------|------------------------------|----------------|--------------|
| 1 | PE_vs_PI | 389 | 191 | 198 |
| 2 | PE_vs_ST | 55 | 22 | 33 |
| 3 | FP_vs_SE | 285 | 240 | 45 |
| 4 | FP_vs_ST | 141 | 89 | 52 |
| 5 | SE_vs_PI | 453 | 173 | 280 |
| 6 | FP_vs_PE | 188 | 124 | 64 |
| 7 | FP_vs_PI | 392 | 247 | 145 |
| 8 | SE_vs_PE | 217 | 60 | 157 |
| 9 | ST_vs_PI | 377 | 189 | 188 |
| 10 | SE_vs_ST | 242 | 46 | 196 |

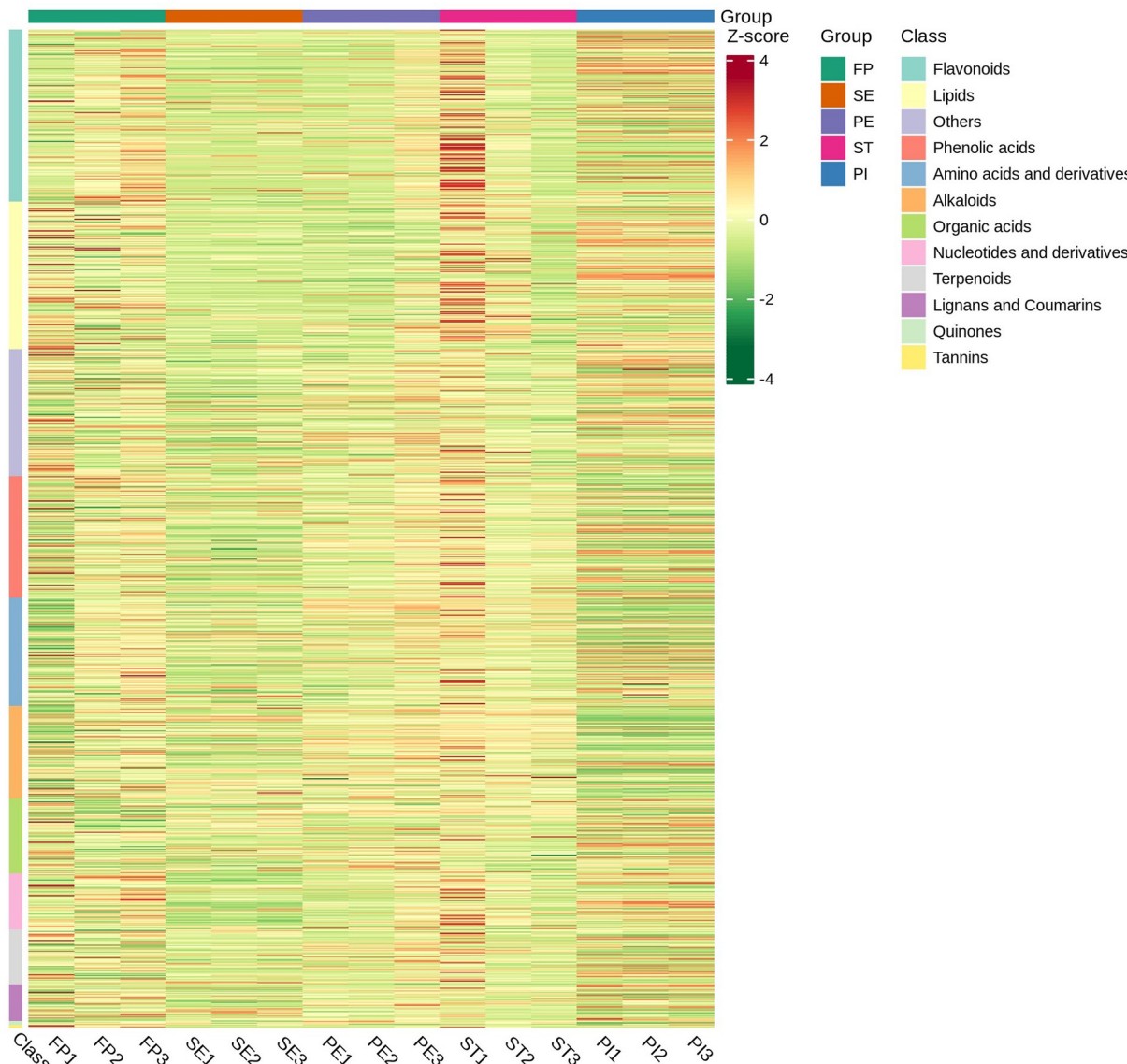

**Fig 2. Sample population clustering plot.** Horizontal is the name of the sample, longitudinal is metabolite information, and different colors are filled with different values obtained after standardized treatment of additional relative content (red represents high content, green represents low content), heat map classified by substance, Class is the first-level classification of the substance.

samples indicate more excellent data stability. In QC samples, over 85% of substances have CV values below 0.5, indicating high stability. Additionally, over 75% of substances in QC samples have CV values below 0.3, indicating very stable experimental data. Based on our CV analysis, as shown in Fig 4, our results showed excellent stability.

## Metabolome data quality assessment via principal component analysis (PCA)

Principal component analysis (PCA) of samples, including quality control (QC), reveals overall metabolite differences and within-group sample variability. PCA indicates metabolome separation between groups, assessing differences within sample groups [28]. To explore

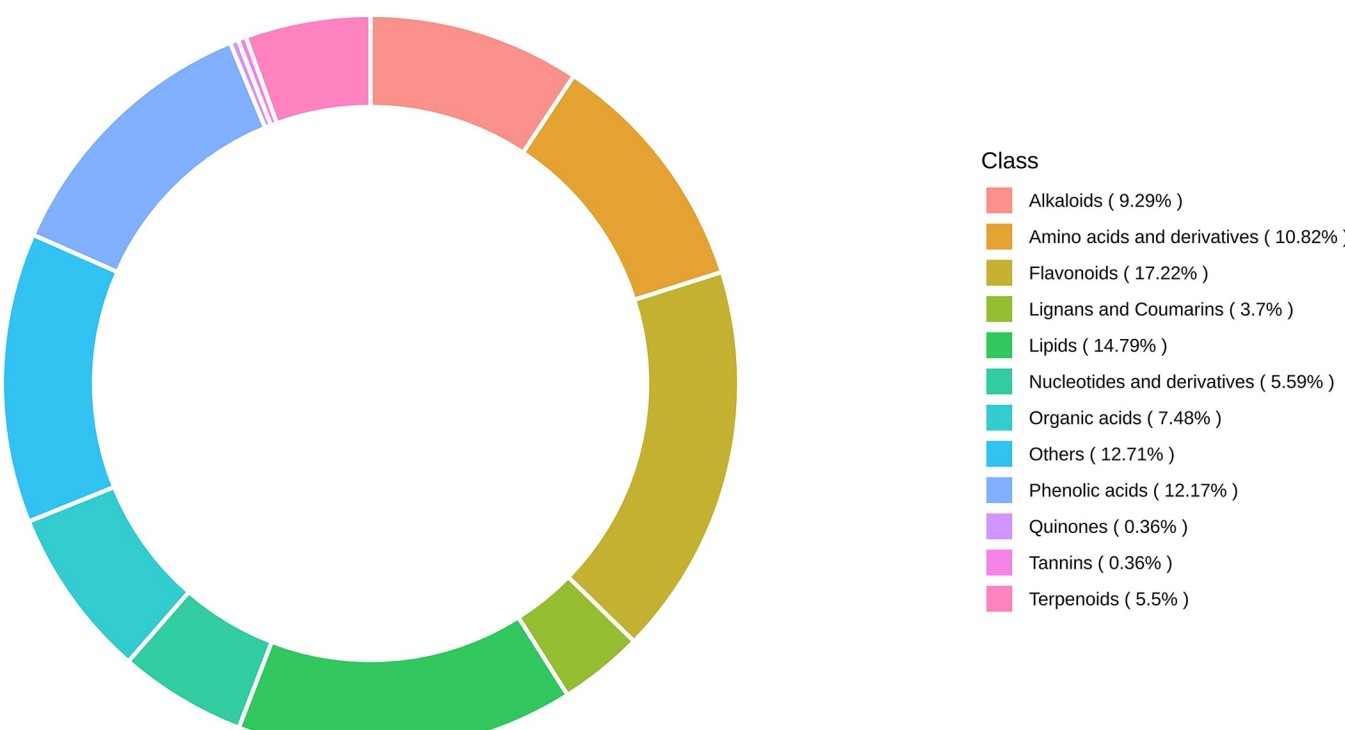

**Fig 3. Metabolite classes are shown in a circular chart.** Each color represents a metabolite class, and the patch area indicates the proportion of the class category.

metabolome dynamics during flower development in each stage, PCA) was conducted using significantly expressed metabolites in all 15 samples, including QC. Results showed FP's similarity to PE and SE, related to non-reproductive flower parts. ST, involved in male reproductive organs, exhibited distinct metabolite expression. PI demonstrated significant variance, suggesting diverse metabolite expression during female reproductive organ development. A 2D PCA plot is shown in Fig 5, where the comparison of the groups and quality control are provided.

## Screening of differentially expressed metabolites

Metabolite expression was screened and analyzed in each developmental stage (group). For group comparison, metabolites with VIP $\geq$ 1 were selected, indicating significant differentiation. Additionally, metabolites with fold change $\geq$ 2 or $\leq$ 2 were chosen. Differences exceeding 5-fold or below 2.0 were considered significant. Detected compounds were categorized as up-regulated, down-regulated, or insignificant. More details are available in S1 File.

## Screening of metabolites between Perianth Primordium (PE) vs Pistil Stage (PI) group

The alteration of the metabolite quantification information in each group was compared by conducting qualitative and quantitative analyses of the identified metabolites, considering the specific sample grouping. The metabolites screened between PE vs PI identified that 455 were significantly enriched, 165 were down-regulated,198 were up-regulated, and 754 were detected but insignificant. A volcano map was plotted to show the up-regulated, down-regulated and insignificant metabolites (Fig 6A). The significantly differentiated metabolites were annotated

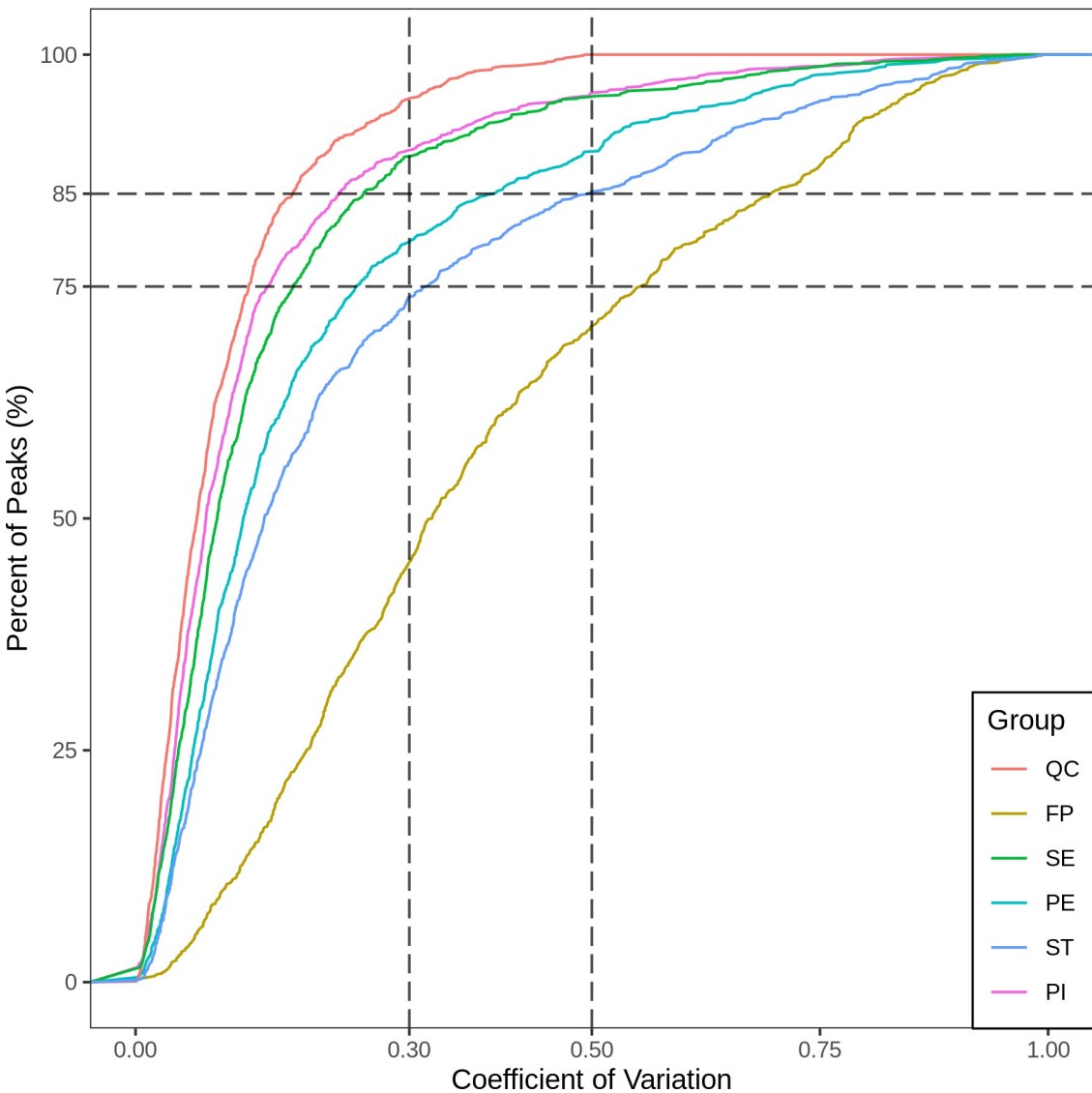

**Fig 4. The coefficient of variation distribution of each sample.** The x-axis represents the coefficient of variation (CV) value. In contrast, the y-axis represents the proportion of substances with a CV value lower than the corresponding value to the total number of substances. Different colors represent different grouped samples; QC is the quality control sample, where the CV values corresponding to the two reference lines perpendicular to the x-axis are 0.3 and 0.5, and the number of substances corresponding to the two reference lines parallel to the x-axis accounts for 75% and 85% of the total number of substances.

to the KEGG database. Valine, leucine, and isoleucine biosynthesis, Isoquinoline alkaloid biosynthesis, and biosynthesis of secondary metabolites were highly enriched, as shown in Fig 6B. Based on the screening criteria; we used Unit Variance Scaling (UV) treatment to the actual relative content of identified differential metabolites to enhance the visual analysis of metabolite content variations. Subsequently, a heat map was generated using the R software package to display the results, as shown in Fig 6C.

## Screening of metabolites between Flower Primordium (FP) vs Perianth Primordium (PE) vs group

A total of 1108 metabolites were detected between the groups FP and PE. We found that 1037 metabolites were insignificant, 48 were up-regulated, and 23 were down-regulated. A volcano

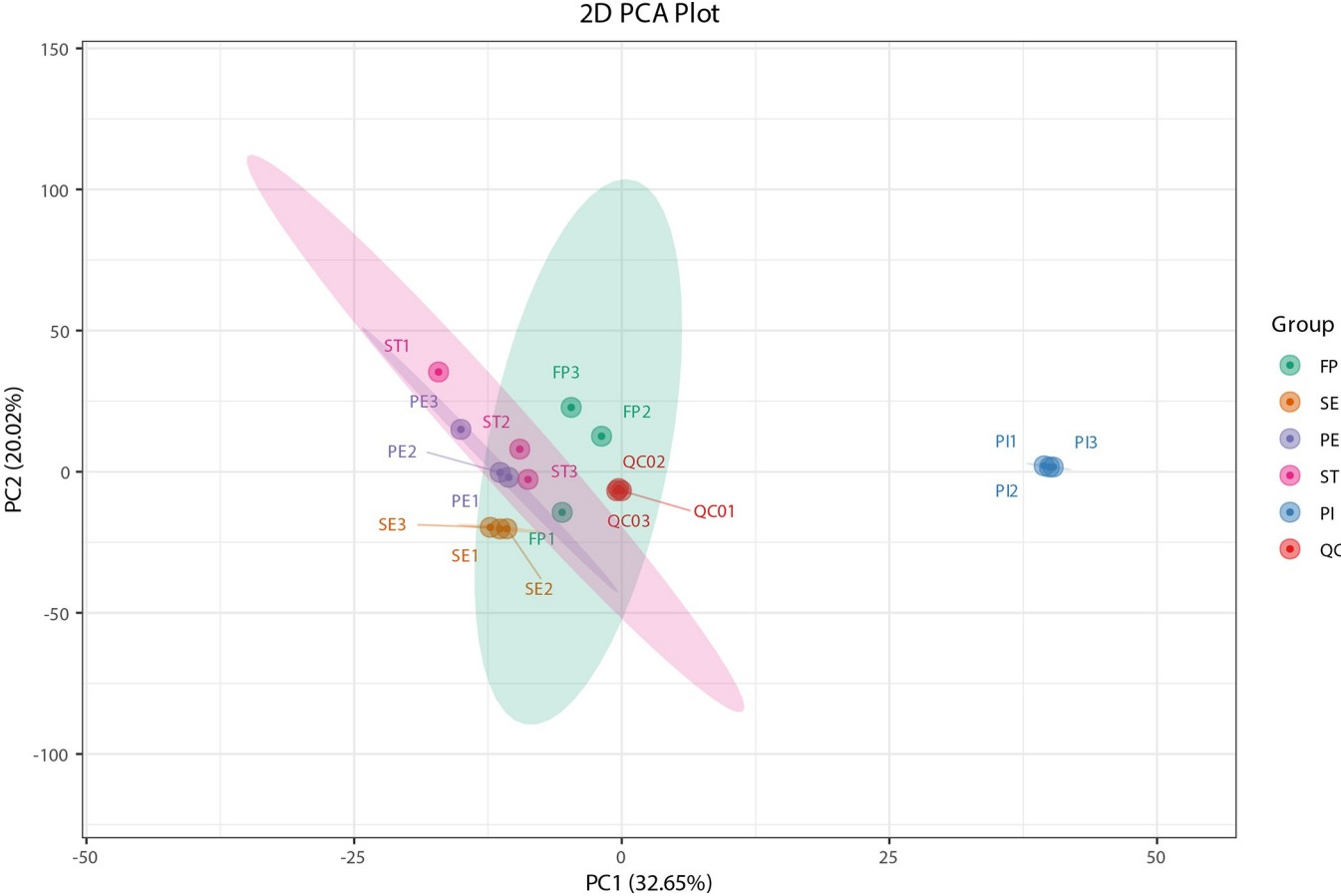

**Fig 5. The PCA score plot of mass spectrometry data for each group of samples, including quality control samples.** PC1 represents the first principal component, PC2 represents the second principal component, PC3 represents the third principal component, and the percentage represents the interpretation rate of the principal component to the data set; each dot in the plot represents a sample, transcripts from the same group are represented by the same color.

plot is provided in Fig 7A. The significantly enriched metabolites were annotated to the KEGG database. Biosynthesis of secondary metabolites, Linoleic acid metabolism, Purine metabolism etc. were highly enriched as shown in Fig 7B. Based on the screening criteria; we used Unit Variance Scaling (UV) treatment to the original relative content of identified differential metabolites to enhance the visual analysis of metabolite content variations. Subsequently, a heat map was generated using the R software package to display the results, as shown in Fig 7C.

## Screening of metabolites between Flower Primordium (FP) vs Sepal Stage (SE) vs group

A total of 1108 metabolites were detected between the groups FP and SE. We found that 1038 metabolites were insignificant; 29 were up-regulated, while 41 were down-regulated. A volcano plot is provided in Fig 8A. The significantly enriched metabolites were annotated to the KEGG database. Flavonoid biosynthesis, Flavone, and flavonol biosynthesis, ABC transporters etc. were highly enriched as shown in Fig 8B. We applied Unit Variance Scaling (UV) treatment to the original relative content of identified differential metabolites based on the screening criteria to enhance the visual analysis of metabolite content variations. Subsequently, a heat map was generated using the R software package to display the results, in Fig 8C.

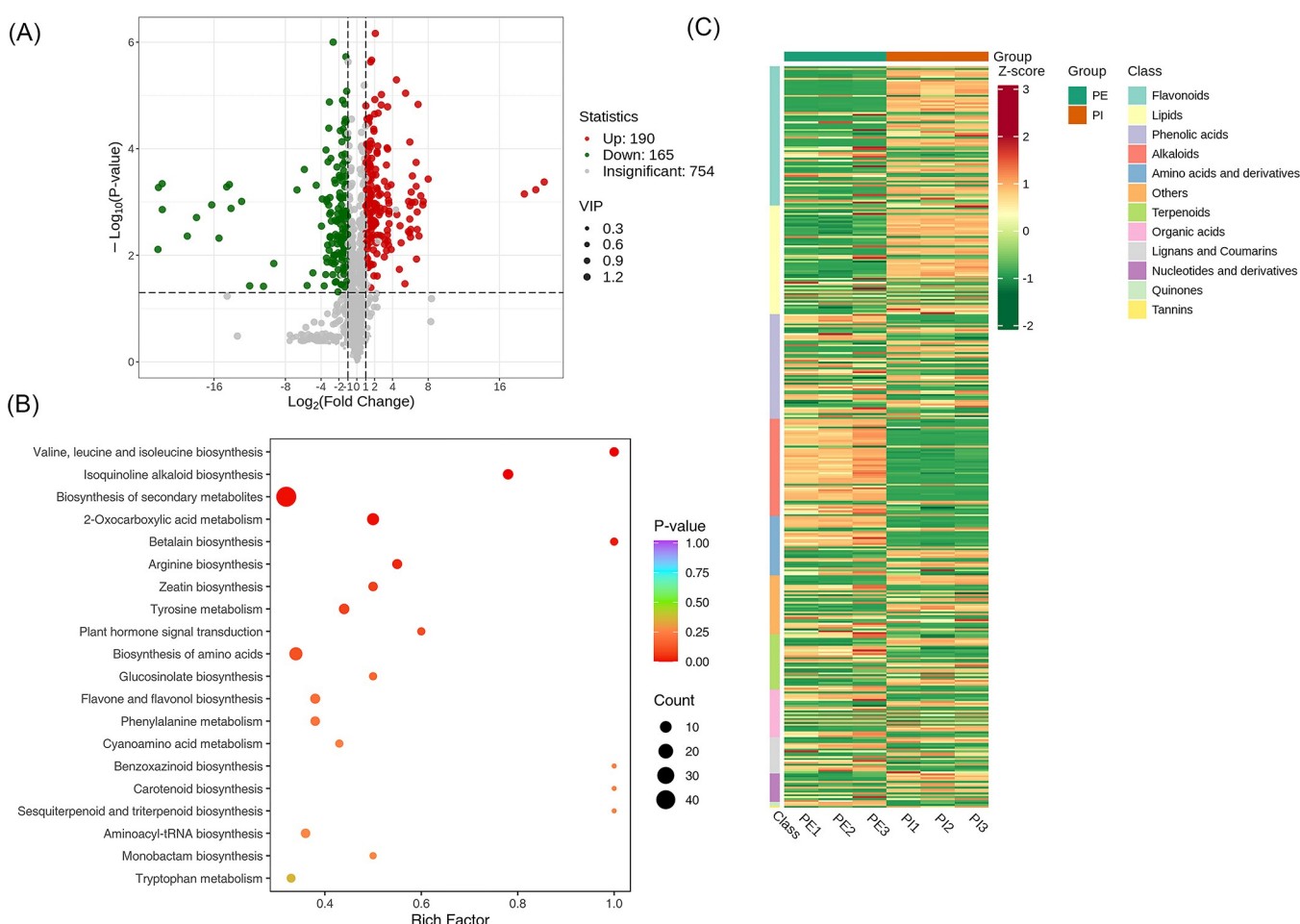

**Fig 6. Qualitative and quantitative analyses of the identified metabolites between PE vs PI group.** (A) Volcano plot of the detected metabolites between PE vs PI. Each dot represents a metabolite. The green dot indicates a down-regulated differential metabolite, the red dot represents an up-regulated differential metabolite, and the gray dot represents nonsignificant metabolites. The x-axis represents the logarithmic value of the fold change in relative content between two sample groups ($log_2FC$). The y-axis represents the level of difference significance ($-log_{10}$P-value), and the dot size represents the VIP value. The volcano plot applies the triple filter conditions of VIP + FC + P-value. (B). The significantly differentiated metabolites were annotated to the KEGG database. The x-axis represents the Rich Factor associated, while the y-axis displays the pathway names arranged in order of their P-values. Each data point's color reflects the P-value's magnitude, with redder colors indicating more significant enrichment. The size of the dots represents the number of differential metabolites that are enriched within each pathway. (C). Heat map of the significantly differentiated metabolites. The sample name is depicted on the horizontal axis, while the vertical axis represents information about differential metabolites. The "Group" parameter describes the group names, and various colors indicate values obtained after standardizing different relative contents. High content is in red, while low content is in green. The heat map is organized based on substances, with "Class" indicating the first-level classification of these substances.

## Screening of metabolites between Flower Primordium (FP) vs Stamen Stage (ST) vs group

A total of 1108 metabolites were detected between the groups FP and ST. We found that 1068 metabolites were insignificant; 25 were up-regulated while 15 were down-regulated. A volcano plot is provided in Fig 9A. The significantly enriched metabolites were annotated to the KEGG database. Isoquinoline alkaloid biosynthesis, Sulfur metabolism, Tyrosine metabolism etc., were highly enriched as shown in Fig 9B. We applied Unit Variance Scaling (UV) treatment to the actual relative content of identified differential metabolites based on the screening criteria to enhance the visual analysis of metabolite content variations. Subsequently, a heat map was generated using the R software package to display the results, Fig 9C.

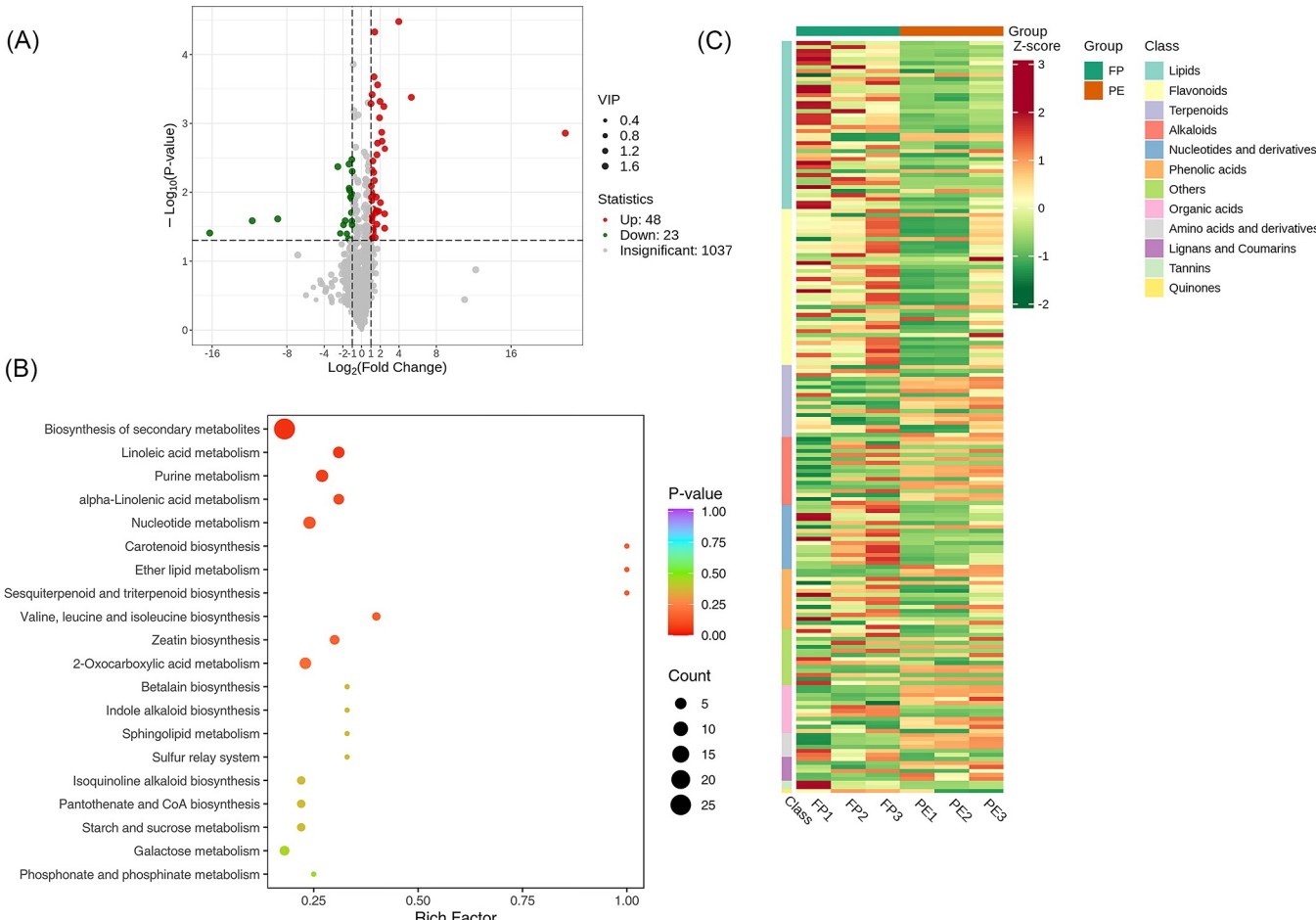

**Fig 7. Qualitative and quantitative analyses of the identified metabolites between FP vs PE groups.** (A) Volcano plot of the detected metabolites between PE vs PI. In the volcano plot, each dot represents a metabolite. The green dot indicates a down-regulated differential metabolite, the red dot represents an up-regulated differential metabolite, and the gray dot represents a detected metabolite that is not significantly different. The x-axis represents the logarithmic value of the fold change in relative content between two sample groups ($\log_2$FC). A larger absolute value on the x-axis indicates a more significant difference in the relative content of the metabolite between the two groups. The y-axis represents the level of difference significance ($-\log_{10}$P-value), and the dot size represents the VIP value. The volcano plot applies the triple filter conditions of VIP + FC + P-value. (B). The significantly differentiated metabolites were annotated to the KEGG database. The x-axis represents the Rich Factor associated with each pathway, while the y-axis displays the pathway names arranged in order of their P-values. Each data point's color reflects the P-value's magnitude, with redder colors indicating more significant enrichment. The size of the dots represents the number of differential metabolites that are enriched within each pathway. (C). Heat map of the significantly differentiated metabolites. The sample name is depicted on the horizontal axis, while the vertical axis represents information about differential metabolites. The "Group" parameter describes the group names, and various colors indicate values obtained after standardizing different relative contents. High content is in red, while low content is in green. The heat map is organized based on substances, with "Class" indicating the first-level classification of these substances.

## Screening of metabolites between Perianth Primordium (PE) vs Stamen Stage (ST) group

A total of 1108 metabolites were detected between the groups FP and SE. We found that 1083 metabolites were insignificant, 10 were up-regulated while 15 were down-regulated. A volcano plot is provided in Fig 10A. The significantly enriched metabolites were annotated to the KEGG database. Phenylpropanoid biosynthesis, Tyrosine metabolism, Carotenoid biosynthesis etc. were highly enriched, Fig 11B. We applied Unit Variance Scaling (UV) treatment to the actual relative content of identified differential metabolites based on the screening criteria to enhance the visual analysis of metabolite content variations. Subsequently, a heat map was generated using the R software package to display the results, Fig 10C.

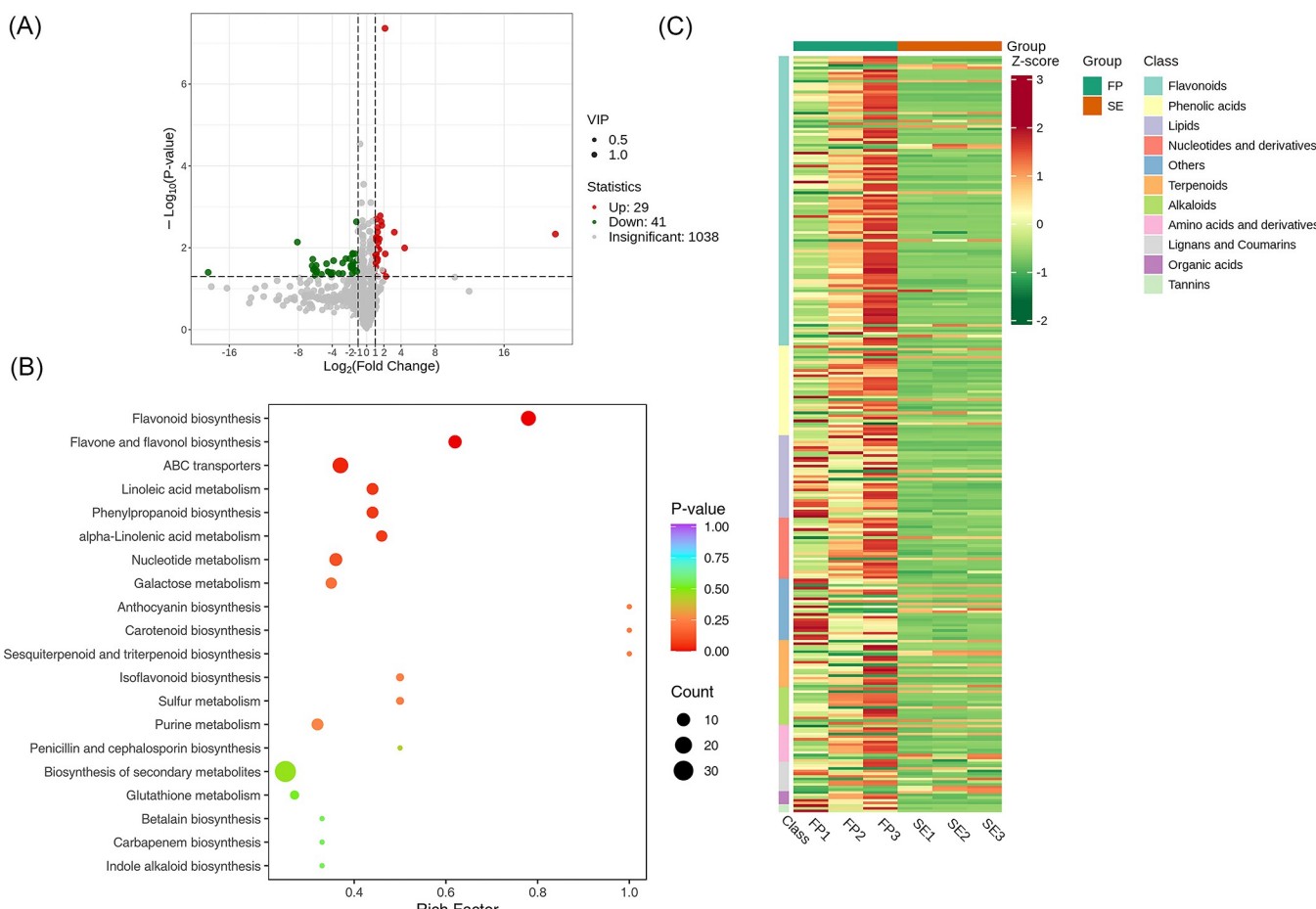

**Fig 8. Qualitative and quantitative analyses of the identified metabolites between FP vs SE groups.** (A) Volcano plot of the detected metabolites between PE vs PI. In the volcano plot, each dot represents a metabolite. The green dot indicates a down-regulated differential metabolite, the red dot represents an up-regulated differential metabolite, and the gray dot represents a detected metabolite that is not significantly different. The x-axis represents the logarithmic value of the fold change in relative content between two sample groups (log₂FC). A more considerable absolute value on the x-axis indicates a more significant difference in the relative content of the metabolite between the two groups. The y-axis represents the level of difference significance (-log₁₀ P-value), and the dot size represents the VIP value. The volcano plot applies the triple filter conditions of VIP + FC + P-value. (B). The significantly differentiated metabolites were annotated to the KEGG database. The x-axis represents the Rich Factor associated with each pathway, while the y-axis displays the pathway names arranged in order of their P-values. Each data point's color reflects the P-value's magnitude, with redder colors indicating more significant enrichment. The size of the dots represents the number of differential metabolites enriched within each pathway. (C). Heat map of the significantly differentiated metabolites. The sample name is depicted on the horizontal axis, while the vertical axis represents information about differential metabolites. The "Group" parameter represents the group names, and various colors indicate values obtained after standardizing different relative contents. High content is in red, while low content is shown in green. The heat map is organized based on substances, with "Class" indicating the first-level classification of these substances.

## Screening of metabolites between Sepal Stage (SE) vs Perianth Primordium (PE) group

We found that 116 metabolites were up-regulated while 55 were down-regulated. A volcano plot is provided in Fig 11A. The significantly enriched metabolites were annotated to the KEGG database. Plant hormone signal transduction, Purine metabolism, Phosphonate and phosphinate metabolism, Sulfur metabolism etc., were highly enriched, Fig 11B. We applied Unit Variance Scaling (UV) treatment to the original relative content of identified differential metabolites based on the screening criteria to enhance the visual analysis of metabolite content variations. Subsequently, a heat map was generated using the R software package to display the results, Fig 11C.

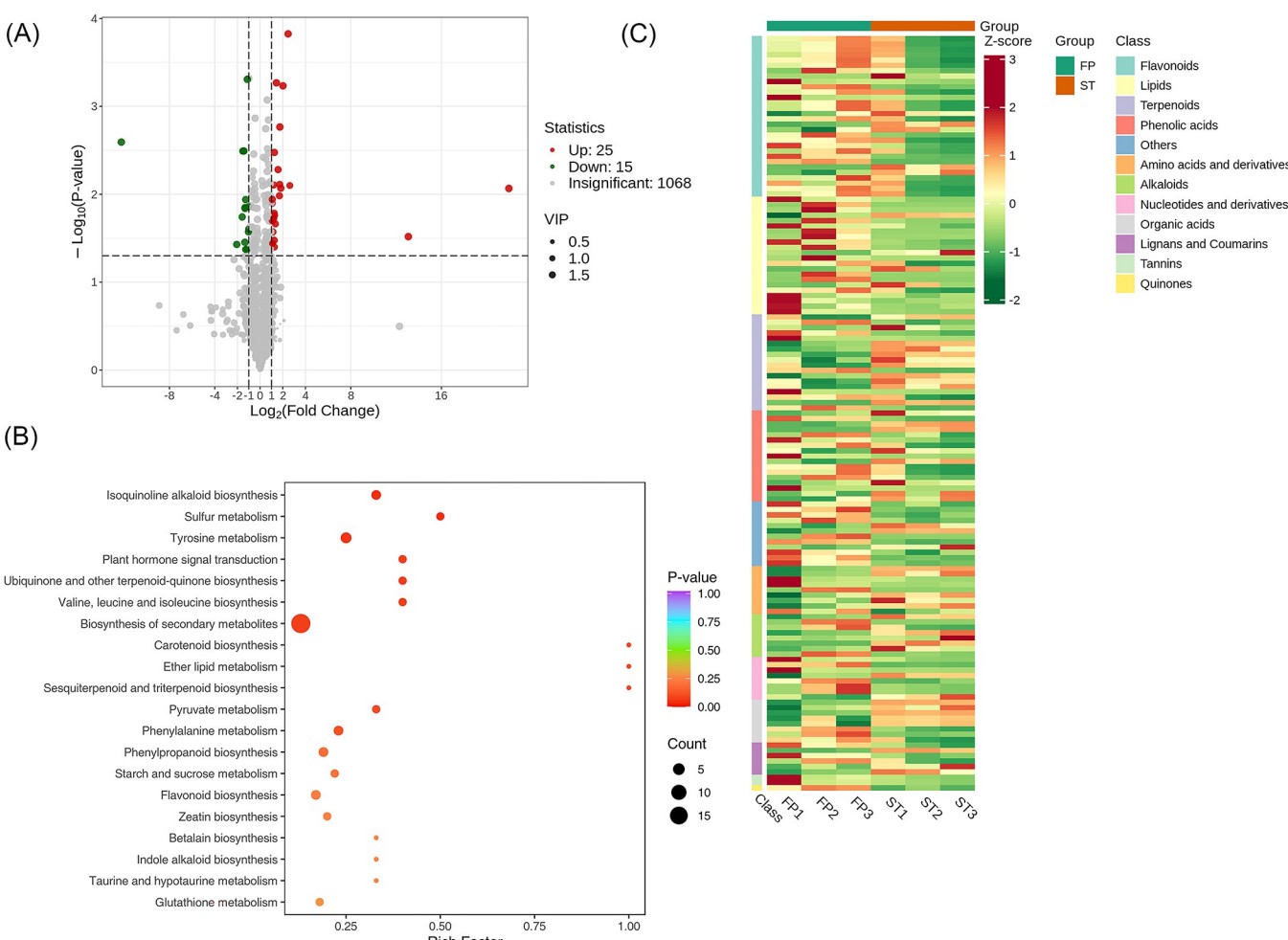

**Fig 9. Qualitative and quantitative analyses of the identified metabolites between FP vs ST groups.** (A) Volcano plot of the detected metabolites between PE vs PI. In the volcano plot, each dot represents a metabolite. The green dot indicates a down-regulated differential metabolite, the red dot represents an up-regulated differential metabolite, and the gray dot represents a detected metabolite that is not significantly different. The x-axis represents the logarithmic value of the fold change in relative content between two sample groups (log₂FC). A larger absolute value on the x-axis indicates a more significant difference in the relative content of the metabolite between the two groups. The y-axis represents the level of difference significance (-log₁₀ P-value), and the dot size represents the VIP value. The volcano plot applies the triple filter conditions of VIP + FC + P-value. (B). The significantly differentiated metabolites were annotated to the KEGG database. The x-axis represents the Rich Factor associated with each pathway, while the y-axis displays the pathway names arranged in order of their P-values. Each data point's color reflects the P-value's magnitude, with redder colors indicating more significant enrichment. The size of the dots represents the number of differential metabolites enriched within each pathway. (C). Heat map of the significantly differentiated metabolites. The sample name is depicted on the horizontal axis, while the vertical axis represents information about differential metabolites. The "Group" parameter represents the group names, and various colors indicate values obtained after standardizing different relative contents. High content is in red, while low content is in green. The heat map is organized based on substances, with "Class" indicating the first-level classification of these substances.

## Screening of metabolites between the Sepal Stage (SE) vs Pistil Stage (PI) group

A total of 1108 metabolites were detected between the groups FP and SE. We found that 668 metabolites were insignificant, 274 were up-regulated, and 166 were down-regulated. A volcano plot is provided in Fig 12A. The significantly enriched metabolites were annotated to the KEGG database. Flavone and flavonol biosynthesis, Flavonoid biosynthesis, Plant hormone signal transduction etc., were highly enriched, Fig 12B. A detailed description is provided in S2 File. We applied Unit Variance Scaling (UV) treatment to the original relative content of identified differential metabolites based on the screening criteria to enhance the visual analysis of

(A)

(B)

(C)

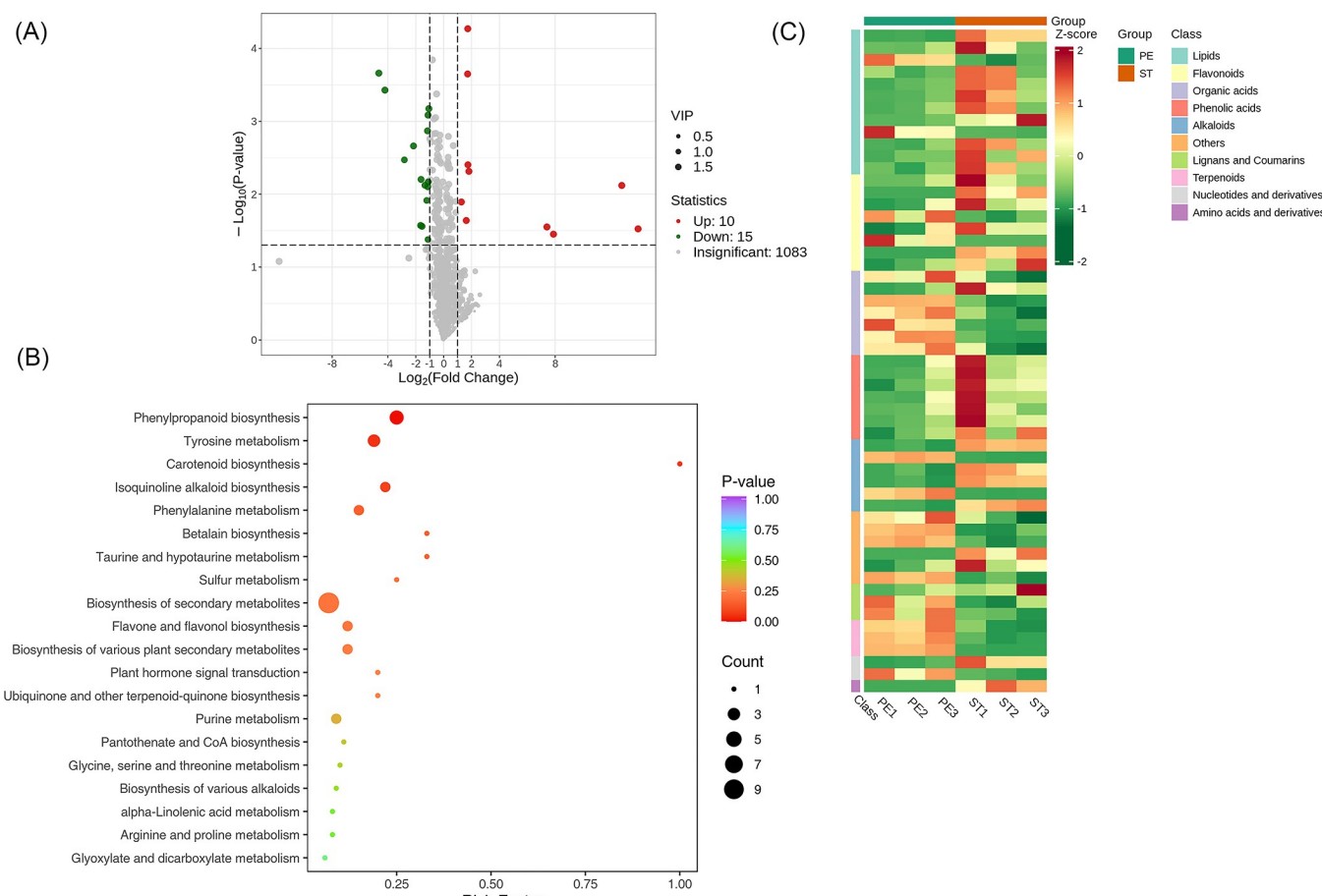

**Fig 10. Qualitative and quantitative analyses of the identified metabolites between PE vs ST groups.** (A) Volcano plot of the detected metabolites between PE vs PI. In the volcano plot, each dot represents a metabolite. The green dot indicates a down-regulated differential metabolite, the red dot represents an up-regulated differential metabolite, and the gray dot represents a detected metabolite that is not significantly different. The x-axis represents the logarithmic value of the fold change in relative content between two sample groups ($\log_2 FC$). A larger absolute value on the x-axis indicates a more significant difference in the relative content of the metabolite between the two groups. The y-axis represents the level of difference significance (-$\log_{10}$ P-value), and the dot size represents the VIP value. The volcano plot applies the triple filter conditions of VIP + FC + P-value. (B). The significantly differentiated metabolites were annotated to the KEGG database. The x-axis represents the Rich Factor associated with each pathway, while the y-axis displays the pathway names arranged in order of their P-values. Each data point's color reflects the P-value's magnitude, with redder colors indicating more significant enrichment. The size of the dots represents the number of differential metabolites that are enriched within each pathway. (C). Heat map of the significantly differentiated metabolites. The sample name is depicted on the horizontal axis, while the vertical axis represents information about differential metabolites. The "Group" parameter represents the group names, and various colors indicate values obtained after standardizing different relative contents. High content is in red, while low content is in green. The heat map is organized based on substances, with "Class" indicating the first-level classification of these substances.

metabolite content variations. Subsequently, a heat map was generated using the R software package to display the results, Fig 12C.

## Screening of metabolites between Sepal Stage (SE) vs Stamen Stage (ST) group

A total of 1108 metabolites were detected between the groups SE and ST. We found that 980 metabolites were insignificant; 84 were up-regulated, while 44 were down-regulated. A volcano plot is provided in Fig 13A. The significantly enriched metabolites were annotated to the KEGG database. Phenylpropanoid biosynthesis, Purine metabolism, Ubiquinone and other terpenoid–quinone biosynthesis etc., were highly enriched, Fig 13B. We applied Unit Variance Scaling (UV) treatment to the original relative content of identified differential metabolites

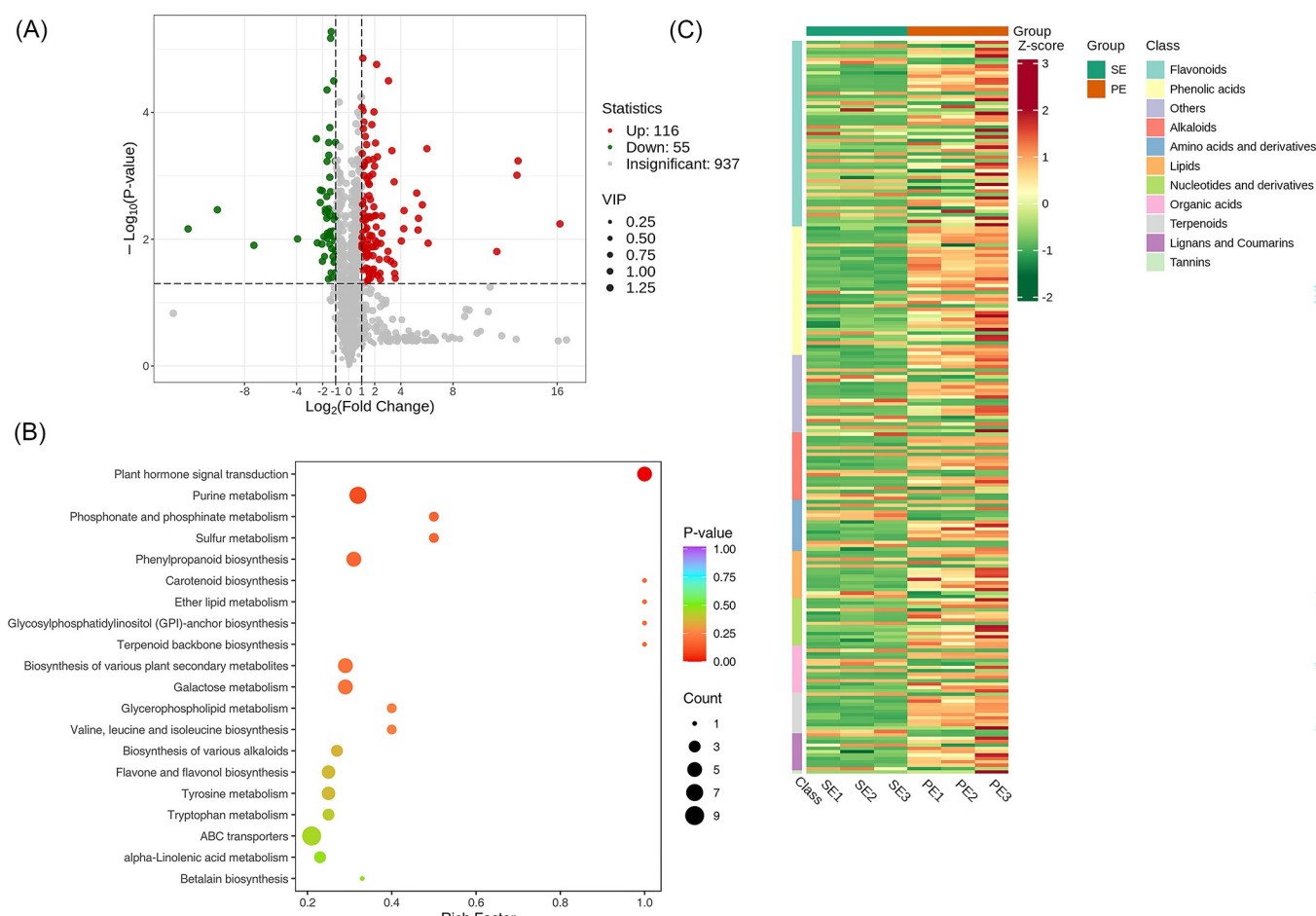

**Fig 11. Qualitative and quantitative analyses of the identified metabolites between PE vs SE groups.** (A) Volcano plot of the detected metabolites between PE vs PI. In the volcano plot, each dot represents a metabolite. The green dot indicates a down-regulated differential metabolite, the red dot represents an up-regulated differential metabolite, and the gray dot represents a detected metabolite that is not significantly different. The x-axis represents the logarithmic value of the fold change in relative content between two sample groups (log₂FC). A larger absolute value on the x-axis indicates a more significant difference in the relative content of the metabolite between the two groups. The y-axis represents the level of difference significance (-log₁₀P-value), and the dot size represents the VIP value. The volcano plot applies the triple filter conditions of VIP + FC + P-value. (B). The significantly differentiated metabolites were annotated to the KEGG database. The x-axis represents the Rich Factor associated with each pathway, while the y-axis displays the pathway names arranged in order of their P-values. Each data point's color reflects the P-value's magnitude, with redder colors indicating more significant enrichment. The size of the dots represents the number of differential metabolites that are enriched within each pathway. (C). Heat map of the significantly differentiated metabolites. The sample name is depicted on the horizontal axis, while the vertical axis represents information about differential metabolites. The "Group" parameter represents the group names, and various colors indicate values obtained after standardizing different relative contents. High content is in red, while low content is in green. The heat map is organized based on substances, with "Class" indicating the first-level classification of these substances.

based on the screening criteria to enhance the visual analysis of metabolite content variations. Subsequently, a heat map was generated using the R software package to display the results, Fig 13C.

## Screening of metabolites between Stamen Stage (ST) vs Pistil Stage (PI) group

A total of 1108 metabolites were detected between the groups FP and SE. We found that 794 metabolites were insignificant, 184 were up-regulated and 131 were down-regulated. A volcano plot is provided in Fig 14A. The significantly enriched metabolites were annotated to the KEGG database. Valine, leucine, and isoleucine biosynthesis, 2−Oxocarboxylic acid

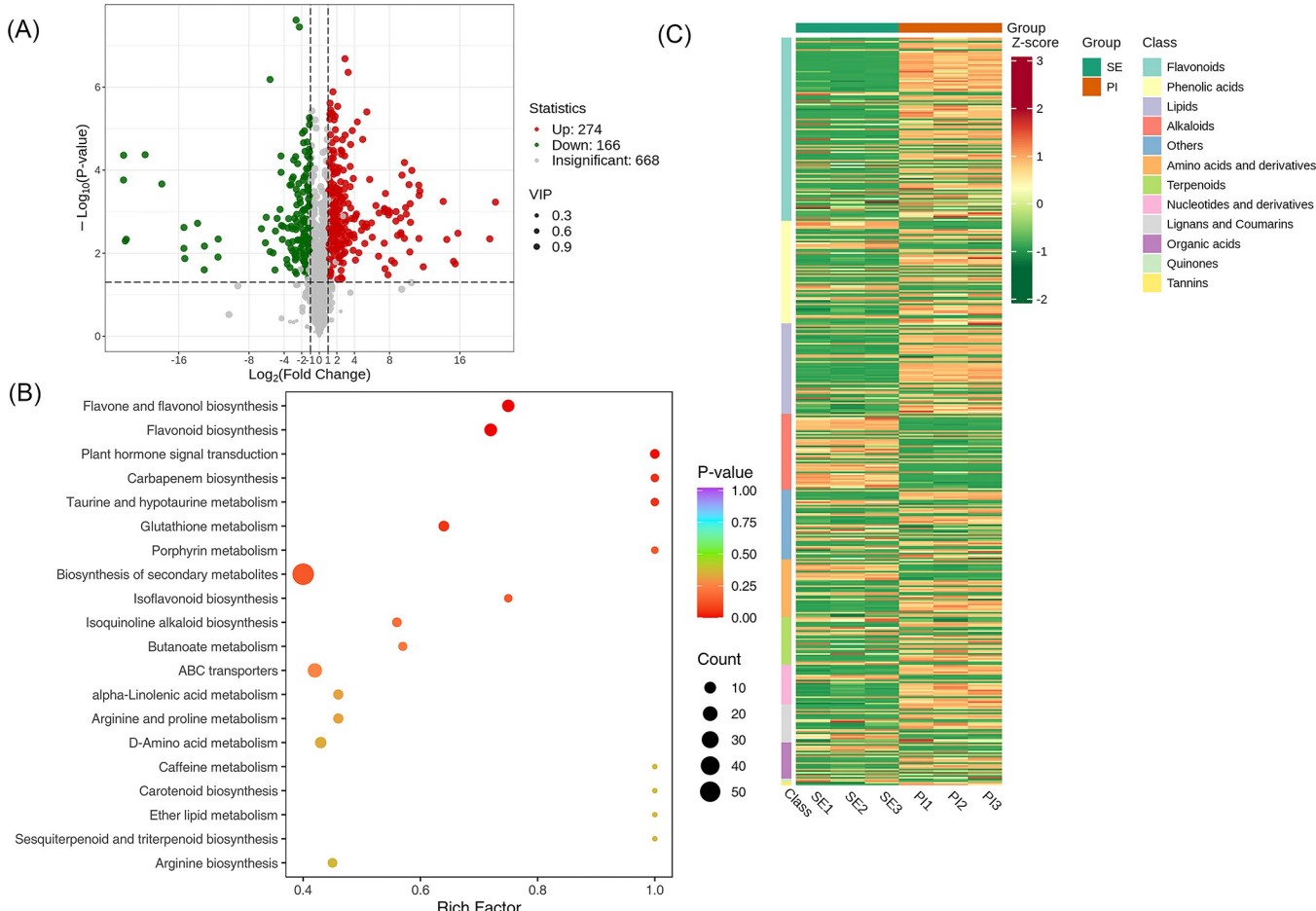

**Fig 12. Qualitative and quantitative analyses of the identified metabolites between PI vs SE groups.** (A) Volcano plot of the detected metabolites between PE vs PI. In the volcano plot, each dot represents a metabolite. The green dot indicates a down-regulated differential metabolite, the red dot represents an up-regulated differential metabolite, and the gray dot represents a detected metabolite that is not significantly different. The x-axis represents the logarithmic value of the fold change in relative content between two sample groups (log$_2$FC). A larger absolute value on the x-axis indicates a more significant difference in the relative content of the metabolite between the two groups. The y-axis represents the level of difference significance (-log$_{10}$P-value), and the dot size represents the VIP value. The volcano plot applies the triple filter conditions of VIP + FC + P-value. (B). The significantly differentiated metabolites were annotated to the KEGG database. The x-axis represents the Rich Factor associated with each pathway, while the y-axis displays the pathway names arranged in order of their P-values. Each data point's color reflects the P-value's magnitude, with redder colors indicating more significant enrichment. The size of the dots represents the number of differential metabolites that are enriched within each pathway. (C). Heat map of the significantly differentiated metabolites. The sample name is depicted on the horizontal axis, while the vertical axis represents information about differential metabolites. The "Group" parameter represents the group names, and various colors indicate values obtained after standardizing different relative contents. High content is in red, while low content is in green. The heat map is organized based on substances, with "Class" indicating the first-level classification of these substances.

metabolism, Arginine biosynthesis etc., were highly enriched, Fig 14B. We applied Unit Variance Scaling (UV) treatment to the original relative content of identified differential metabolites based on the screening criteria to enhance the visual analysis of metabolite content variations. Subsequently, a heat map was generated using the R software package to display the results, Fig 14C.

## Key findings in metabolites across developmental stages

In this study, a detailed analysis of various metabolites was conducted across the different developmental stages of *Adonis amurensis* flowers. Among the key metabolites identified, flavonoids were particularly significant. Notably, compounds such as Tangeretin (a

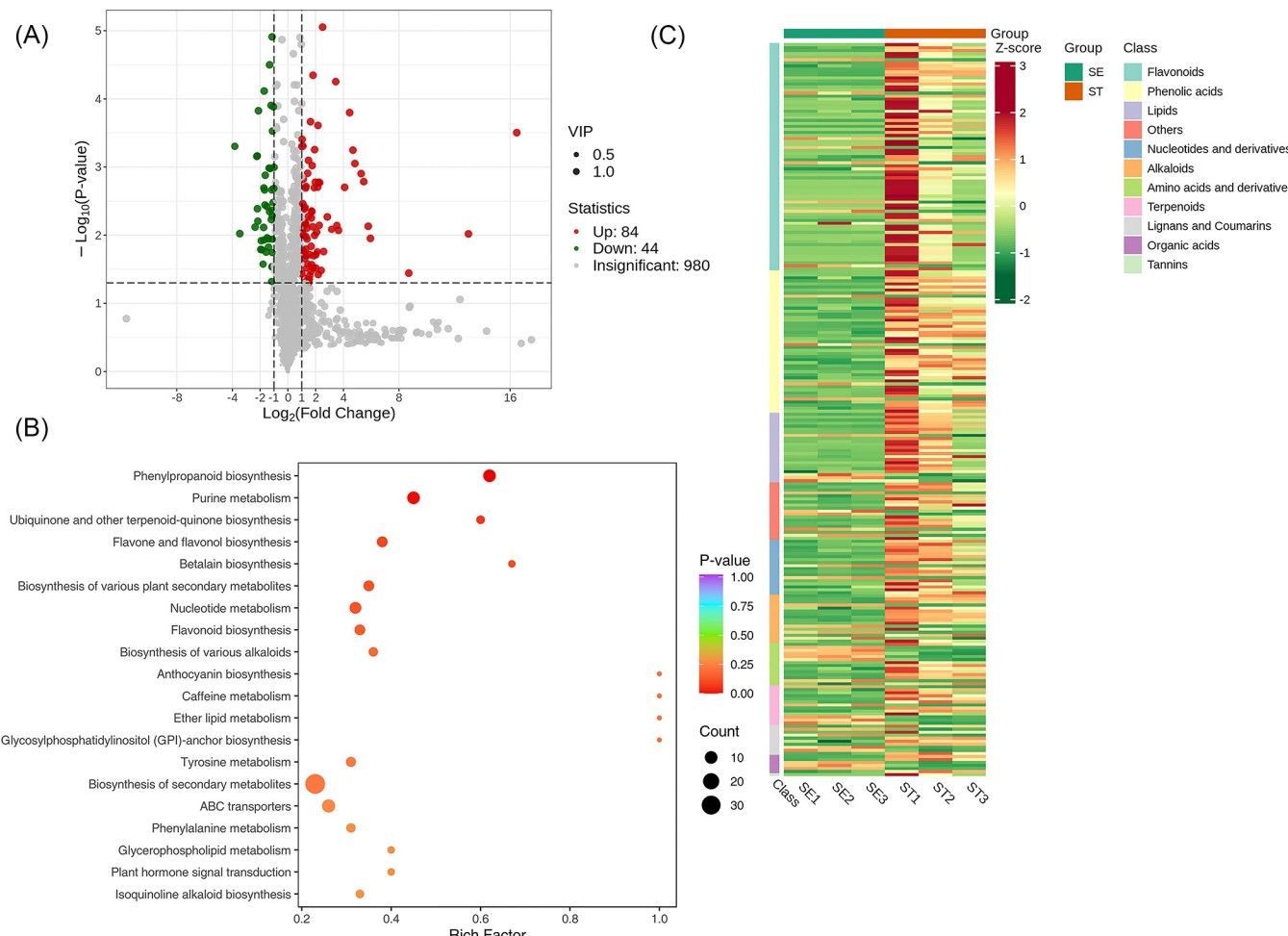

**Fig 13. Qualitative and quantitative analyses of the identified metabolites between ST vs SE groups.** (A) Volcano plot of the detected metabolites between PE vs PI. In the volcano plot, each dot represents a metabolite. The green dot indicates a down-regulated differential metabolite, the red dot represents an up-regulated differential metabolite, and the gray dot represents a detected metabolite that is not significantly different. The x-axis represents the logarithmic value of the fold change in relative content between two sample groups (log$_2$FC). A larger absolute value on the x-axis indicates a greater difference in the relative content of the metabolite between the two groups. The y-axis represents the level of difference significance (-log$_{10}$P-value), and the dot size represents the VIP value. The volcano plot applies the triple filter conditions of VIP + FC + P-value. (B). The significantly differentiated metabolites were annotated to the KEGG database. The x-axis represents the Rich Factor associated with each pathway, while the y-axis displays the pathway names arranged in order of their P-values. Each data point's color reflects the P-value's magnitude, with redder colors indicating more significant enrichment. The size of the dots represents the number of differential metabolites enriched within each pathway. (C). Heat map of the significantly differentiated metabolites. The sample name is depicted on the horizontal axis, while the vertical axis represents information about differential metabolites. The "Group" parameter represents the group names, and various colors indicate values obtained after standardizing different relative contents. High content is in red, while low content is in green. The heat map is organized based on substances, with "Class" indicating the first-level classification of these substances.

pentamethoxyflavone) were observed, which play essential roles in plant defense and possess potential pharmacological properties. The pathway of flavonoid biosynthesis was highly enriched, especially between reproductive stages like the Stamen Stage and Pistil Stage. These compounds are critical for flower coloration, which helps in attracting pollinators, and they also offer UV protection to the reproductive organs.

Another important group of metabolites identified was amino acids and their derivatives. For example, N-Acetyl-L-threonine was detected, showing upregulation during specific developmental stages. Additionally, amino acids such as valine, leucine, and isoleucine were prominent, which underscores their role in protein synthesis and the development of flower organs.

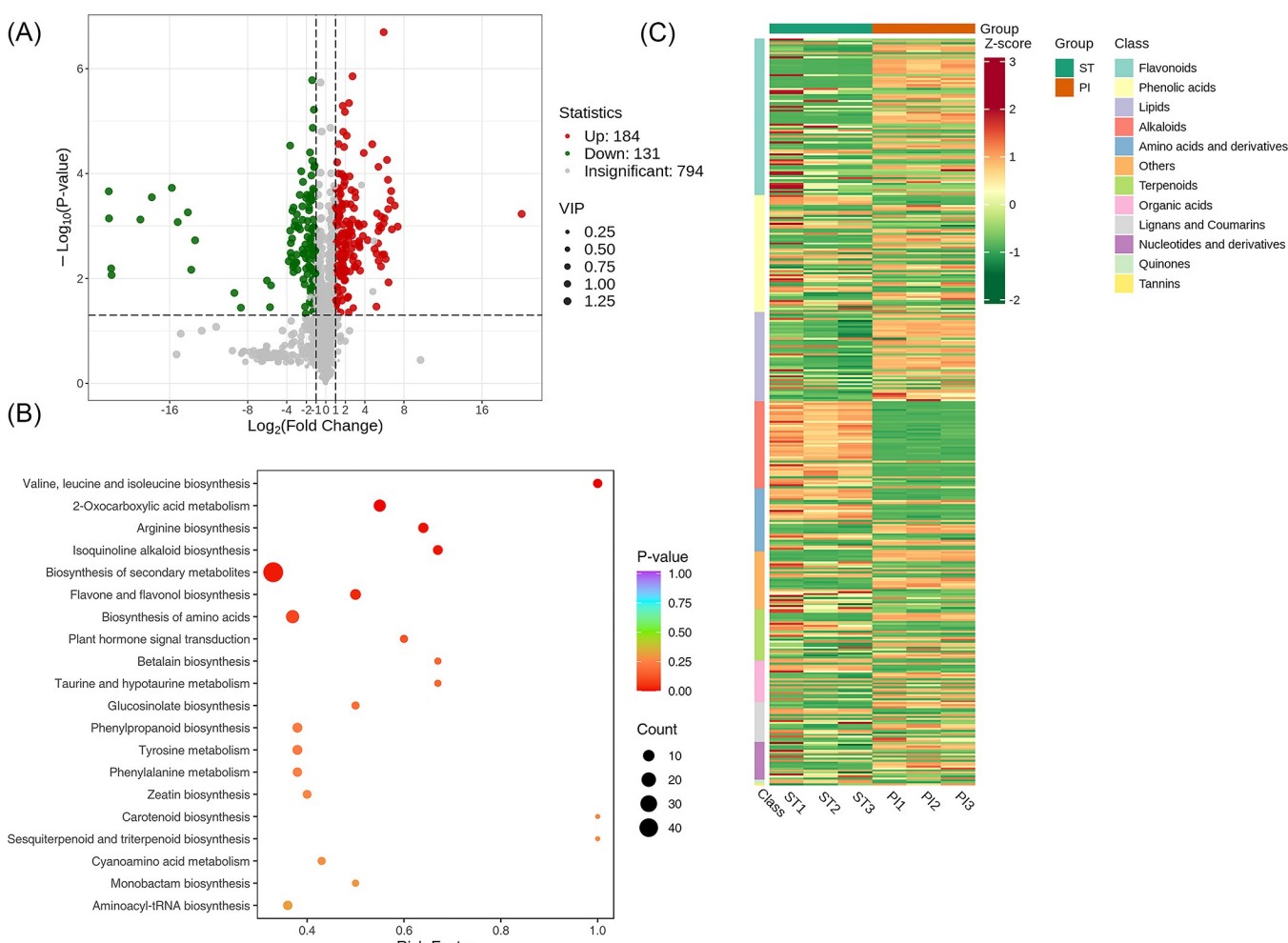

**Fig 14. Qualitative and quantitative analyses of the identified metabolites between ST vs PI groups.** (A) Volcano plot of the detected metabolites between PE vs PI. In the volcano plot, each dot represents a metabolite. The green dot indicates a down-regulated differential metabolite, the red dot represents an up-regulated differential metabolite, and the gray dot represents a detected metabolite that is not significantly different. The x-axis represents the logarithmic value of the fold change in relative content between two sample groups (log$_2$FC). A larger absolute value on the x-axis indicates a greater difference in the relative content of the metabolite between the two groups. The y-axis represents the level of difference significance (-log$_{10}$P-value), and the dot size represents the VIP value. The volcano plot applies the triple filter conditions of VIP + FC + P-value. (B). The significantly differentiated metabolites were annotated to the KEGG database. The x-axis represents the rich factor associated with each pathway, while the y-axis displays the pathway names arranged in order of their P-values. Each data point's color reflects the P-value's magnitude, with redder colors indicating more significant enrichment. The size of the dots represents the number of differential metabolites enriched within each pathway. (C). Heat map of the significantly differentiated metabolites. The sample name is depicted on the horizontal axis, while the vertical axis represents information about differential metabolites. The "Group" parameter describes the group names, and various colors indicate values obtained after standardizing different relative contents. High content is in red, while low content is in green. The heat map is organized based on substances, with "Class" indicating the first level classification of these substances.

Enhanced biosynthesis of these amino acids, particularly in the Pistil Stage, suggests increased metabolic activity to support processes like ovule development and maturation.

Phenolic acids were also identified as crucial metabolites, with compounds like 3-O-p-Coumaroylshikimic acid-O-glucoside playing a significant role. These metabolites are well-known for their antioxidant properties, contributing to the plant's defense mechanisms. Phenolic acid pathways were particularly active during the transition to reproductive stages, indicating their role in protecting developing organs and supporting reproductive functions.

The study also highlighted the presence of organic acids such as 3-Isopropylmalic Acid, which serve as intermediates in various metabolic pathways. These organic acids are important

for maintaining cellular pH and act as precursors for other vital compounds. The levels of organic acids varied significantly across different developmental stages, with notable changes during the early phases, reflecting shifts in the flower's metabolic state.

In addition to these groups, several secondary metabolites and specialized compounds were detected. These were linked to pathways like plant hormone signal transduction, which is essential for regulating the growth and development of flower organs. The differential expression of these secondary metabolites, including those associated with flavonoid and phenylpropanoid biosynthesis, suggests a role in ensuring reproductive success and adaptation to environmental conditions.

### Differential expression highlights

A closer look at differential expression revealed that flavonoids and amino acids were significantly altered between the Stamen Stage and the Pistil Stage (ST_vs_PI). This indicates that these compounds play distinct roles during these phases, with some being upregulated to meet the demands of reproductive development. Additionally, comparisons between the Sepal Stage and Stamen Stage (SE_vs_ST) showed notable increases in flavonoid-related metabolites, highlighting their role in the development of reproductive structures.

These findings emphasize a complex interplay of primary and secondary metabolites that not only facilitate flower development but also enhance pollinator attraction and provide essential protection to the plant. By identifying and understanding these key metabolites and their associated pathways, the study offers a deeper insight into the metabolomic shifts occurring during critical developmental stages in *A. amurensis*.

### Discussion

Flowers serve as the reproductive structures of angiosperms, also known as flowering plants. They serve a very crucial role in the life cycle of plants by producing the male and female reproductive organs. The biomass of the flower varies extensively among angiosperms [29, 30]. Flower development can be affected by several factors such as environmental factors, genetic factors etc. At the advent of latest technology many researchers got interested in studying the flowers as molecular levels. Since transcriptome and proteome studies provide deep insights into the expression pattern of genetic materials; metabolomics helps in identifying small molecules, or metabolites, present in a biological system [20, 31]. It is helpful in metabolite profiling during different developmental stages and provides insights into the metabolic pathways and regulatory networks underlying this developmental process. Integrating metabolomic data with transcriptomic and proteomic information offers a holistic understanding of the molecular mechanisms governing flowering in plants [23]. Using metabolomics, several researchers have deeply studied the flowers of different plants in different contexts such as color [32, 33], metabolites of the flowers of transgenic plants [34], antioxidants and anti-cancer molecules from the flowers [35], etc. Here in this study, we have performed the metabolomic analysis of the *A. amurensis* flower at different developmental stages. These stages were "Flower Primordium (FP)", "Sepal Stage (SE)", "Perianth Primordium (PE)", "Stamen Stage (ST)" and "Pistil Stage (PI)". Previously, we have studied the differentiation processes and morphological changes of the *A. amurensis* flower developmental stages [15]. With high confidence, this study utilized ultra-high performance liquid chromatography-tandem mass spectrometry (UPLC-MS/MS) to quantify and analyze metabolites. In our findings, the PCA plot illustrated metabolite expression profiles between non-reproductive (FP, SE, PE) and reproductive (ST, PI) stages distinctively. The significant difference was observed in the reproductive stages, particularly the Pistil Stage (PI), reflects the complex metabolic demands during

ovule development and seed formation [36]. These differences have ecological and physiological implications. Besides attracting pollinators through color and scent, protecting reproductive organs from UV radiation, they ensure successful fertilization and seed maturation [37, 38]. The PCA findings provide a comprehensive understanding of the dynamic metabolic shifts occurring during critical developmental stages, contributing to our knowledge of plant reproductive biology.

Flower Primordium is the early developmental stage of a flower, where the floral organs start to form from meristematic cells [39]. In our research, the groups can rightly be divided into two parts. The reproductive part of the flower consists of Flower primordium (FP), Sepal stage (SE), and Perianth primordium (PE). The reproductive part is the Stamens stage (ST) and the Pistil stage (PT). We found that the Sepal stage (SE) vs Pistil stage (PI) showed the highest number of metabolites expressed. The flower undergoes several developmental stages before it can produce reproductive organs. Each stage is accumulated by a number of metabolites [40, 41]. Both of these stages are very crucial in flower development. Sepals, which are the outermost floral organs, begin to differentiate and take shape during the Sepal stage. Pistil gives rise the ovaries which produces the female reproductive organs. In our findings, we have observed significant metabolic shifts involving pathways such as flavonoid biosynthesis, flavone and flavonol biosynthesis. In addition, there was a significant shift in plant hormone signal transduction. These pathways are involved in flower coloration, growth regulation, and the development of reproductive organs. The flavonoid biosynthesis pathway not only contributes to the pigmentation and visual attraction of the flowers but also provides UV protection and aids in fertility. Studies such as those by Zhao et al. [42] and others [22, 43] have demonstrated the critical functions of flavonoids in flower development and pollinator attraction. The significant differences in metabolite expression between the ST and PI stages underscore the distinct metabolic demands of these developmental phases, the upregulation of metabolites in the PI stage, especially those involved in the biosynthesis of secondary metabolites and plant hormones, highlights the metabolic intensification important for ovule development and ultimately the formation of seed. This phenomenon is strongly supported by findings from Önder et al. [44], who noted the role of secondary metabolites in flower development in *Rosa damascena*. Moreover, the differential expression of metabolites involved in amino acid biosynthesis pathways, like the valine, leucine, and isoleucine biosynthesis pathways detected in our research, suggests an enhanced protein synthesis activity in the pistil to support ovary development and seed maturation. Similar findings were reported by Chakraborty et al. [40], emphasizing the role of amino acids in floral organ development. These findings are strongly supported by the literature [45–47], providing a comparative analysis that highlights how *A. amurensis* adapts its metabolic pathways to environmental cues and developmental needs. The incorporation of metabolomic data with previously reported transcriptomic and metabolomic data offers a complete view of the regulatory mechanisms in flower development.

Flavone and flavanol biosynthesis pathway are the highly enriched KEGG pathway in SE vs PI. This pathway is mainly involved in color pigmentation [42, 48]. Besides, the role of Flavone and flavanol biosynthesis has been demonstrated in flower development [43]. Using High-performance liquid chromatography (HPLC) analysis, researchers found that flavone accumulates from the early flower bud stage [49]. Flavonoid biosynthesis and Plant hormone signal transduction pathways are significantly enriched (S2 File). The biosynthesis of secondary metabolites pathway was enriched considerably between Flower primordium (FP) and Perianth primordium (PE). Secondary metabolites are low molecular weight molecules with diverse chemical structures and biological activities. These secondary metabolites have been reported to play a role in flower development in *Rosa damascena* [44]. Others studied the volatile organic compounds emissions from the *Luculia pinceana* flower and found that these volatile

organic compounds have been changed at different stages of flower development [50]. The PCA plot showed that the non-reproductive parts of the flowers showed similarities in the expression of metabolites while the reproductive parts such as Stamens (ST) and Pistil (PI) showed visible variance. Out results are in agreement with other researchers who showed similar results [51, 52]. The difference in the expression of metabolites between the male (ST) and female organs (PI) showed high variance. In particular we have observed that the expression of metabolites at every developmental stage showed a value of variance. This study provides a first insight into the expression of different metabolites at different developmental stages of *A. amurensis* flower. The significant variations in the concentration and presence of key metabolites found in our study, such as flavonoids and saponins, across different developmental stages of *A. amurensis* flowers; are known for their anti-inflammatory, antioxidant, and cardio-protective properties [53–55]. By finding the stages at which these metabolites are most rich, our study advocates best harvest times that could improve the efficacy and yield of medicinal extracts derived from this plant. For example, if specific anti-cancer or cardio-protective metabolites peak during the Pistil stage, as suggested by our analysis, targeted harvesting during this stage could boost the medical efficacy of extracts used in treatment formulations. This approach is supported by the findings from other pharmacological studies, which have shown the therapeutic consequence of timing in plant metabolite extraction [56]. Understanding the dynamic changes in metabolite profiles during flower development stipulates crucial insights for agricultural practices and the sustainable use of *A. amurensis* as a medicinal resource. This knowledge can model cultivation strategies to enhance yield and quality, while also informing conservation efforts, especially given the plant's status as a valuable but potentially vulnerable medicinal species. Moreover, our research can encourage advancements in phytopharmacology and herbal medicine by detecting candidate compounds for further investigation in drug development processes.

## Conclusion

Our study represents a systematic investigation of the changes in primary and secondary metabolites during the developmental stages of *A. amurensis* flowers. Using UPLC-MS/MS, we identified significant differences in metabolites that correspond to important stages of flower development. Specifically, we observed marked increases in flavonoids such as quercetin and kaempferol during the Pistil Stage (PI), which are essential for reproductive organ development and pollinator attraction. Amino acids such as valine, leucine, and isoleucine were also upregulated, indicating enhanced protein synthesis activity necessary for ovule development and seed maturation. The association between flower development and metabolite changes underscores the complex regulatory networks that drive these processes. Our pathway enrichment analysis revealed that pathways such as flavonoid biosynthesis and plant hormone signal transduction play pivotal roles in flower maturation, contributing to the plant's adaptive and reproductive success.

## Future prospects

To address the gaps identified in this study, future research should focus on several key areas. First, expanding the sample size and incorporating additional biological replicates would enhance the robustness of the findings. Second, targeted quantitative assays for specific metabolites identified in this study could validate and refine the findings of our research. Third, exploring the functional roles of these identified metabolites in other species or under different environmental conditions could provide broader insights into their biological significance. Finally, integrating metabolomic data with transcriptomic and proteomic analyses would offer

a more holistic understanding of the molecular mechanisms underlying flower development in *A. amurensis*. Such integrated approaches could pave the way for novel applications in agriculture, horticulture, and pharmacology, particularly in the development of new therapeutic agents derived from stage-specific metabolites.

## Supporting information

**S1 File. Significant_Metabolites_Summary.**
(XLSX)

**S2 File. SE_vs_PI_KEGG.**
(XLSX)

## Author Contributions

**Conceptualization:** Zun Xingzun, Wang Hongtao.

**Data curation:** Zun Xingzun, Wang Hongtao.

**Formal analysis:** Zun Xingzun, Wang Hongtao.

**Funding acquisition:** Wang Hongtao.

**Investigation:** Zun Xingzun, Wang Hongtao.

**Methodology:** Zun Xingzun, Wang Hongtao.

**Project administration:** Wang Hongtao.

**Resources:** Zun Xingzun, Wang Hongtao.

**Software:** Zun Xingzun, Wang Hongtao.

**Supervision:** Wang Hongtao.

**Validation:** Zun Xingzun, Wang Hongtao.

**Visualization:** Zun Xingzun, Wang Hongtao.

**Writing – original draft:** Zun Xingzun, Wang Hongtao.

**Writing – review & editing:** Wang Hongtao.

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
