## [Decision Letter · Decision Letter 0]

9 Nov 2023

PONE-D-23-24707Metabolomics Revealed a Comprehensive Understanding of the Biochemical and Genetic Mechanisms Underlying the Developmental Shift of the Adonis amurensis Regel et Radde flowersPLOS ONE

Dear Dr. Hongtao,

Thank you for submitting your manuscript to PLOS ONE. After careful consideration, we feel that it has merit but does not fully meet PLOS ONE’s publication criteria as it currently stands. Therefore, we invite you to submit a revised version of the manuscript that addresses the points raised during the review process.

We look forward to receiving your revised manuscript.

Kind regards,

Pankaj Bhardwaj, Ph.D.

Academic Editor

PLOS ONE

Journal Requirements:

Whilst you may use any professional scientific editing service of your choice, PLOS has partnered with both American Journal Experts (AJE) and Editage to provide discounted services to PLOS authors. Both organizations have experience helping authors meet PLOS guidelines and can provide language editing, translation, manuscript formatting, and figure formatting to ensure your manuscript meets our submission guidelines. To take advantage of our partnership with AJE, visit the AJE website (http://aje.com/go/plos) for a 15% discount off AJE services. To take advantage of our partnership with Editage, visit the Editage website (www.editage.com) and enter referral code PLOSEDIT for a 15% discount off Editage services. If the PLOS editorial team finds any language issues in text that either AJE or Editage has edited, the service provider will re-edit the text for free.

This research was supported by “Growing Up Scholar Climbing Program, Changchun University” No. zkp202205.

Reviewers' comments:

Reviewer's Responses to Questions

**Comments to the Author**

1. Is the manuscript technically sound, and do the data support the conclusions?

Reviewer #1: No

Reviewer #2: Yes

2. Has the statistical analysis been performed appropriately and rigorously? 

Reviewer #1: No

Reviewer #2: Yes

3. Have the authors made all data underlying the findings in their manuscript fully available?

Reviewer #1: No

Reviewer #2: Yes

4. Is the manuscript presented in an intelligible fashion and written in standard English?

Reviewer #1: No

Reviewer #2: Yes

5. Review Comments to the Author

Reviewer #1: Dear authors

Adonis amurensis Regel et Radde holds significant importance as a spring ephemeral

plant. It’s medicinal uses have been extensively studied in targeting cardiac

insufficiency and edema. Its anticancer potential has also been identified. Previous

investigation delineated its physical characteristics, flowering patterns, reproductive

and cultural biology and molecular biology. However, the flower developmental stageassociated metabolic changes are poorly understood. This study aimed to investigate

the changes in metabolites and their interactions at five developmental stages, namely

Flower Primordium (FP), Sepal Stage (SE), Perianth Primordium, (PE) Stamens Stage

(SE), and Pistil Stage (PI). Hight performance liquid chromatography coupled with

mass spectroscopy was performed to identify the flower developmental stage

associated metabolites. The analysis of the data was done through MSEA and KEGG

database was employed for metabolite annotation. Flower development stages

significantly impacted both the quantity and quality of metabolite contents, with notable

shifts in the types and amounts of metabolites occurring throughout the flower

development process. Particularly, the SE vs PI stage exhibited the highest number

(453) of significantly differentiated metabolites, indicating a substantial increase in

metabolite expression during the development of reproductive organs. Conversely, the

PE vs ST stage showed the lowest number of significantly differentiated metabolites.

The study provides novel insights into the analysis of distinct metabolite contents at

each stage of floral development. The information obtained will facilitate better

understanding of studying the factors involved in flower development and transition

from one stage to another

Based on the provided abstract and supporting materials. I couldn't access the main draft due to the authors not having uploaded the full text of the article; therefore, I am making my decision as a rejection.

Reviewer #2: The paper entitled Metabolimics revealed a comprehensive understanding of the biochemical and genetic nechanisms underlying the developmental shift of the Adonis amurensis Regel et Raddle flowers pertaining to metabolic analysis at differnet stages of floral development. It has been previously reported in number of plant species including the model plants that there is differential expression of genes during developmental stages which transcribes into differential metabolic profiles. The idea of the work is good and novel and pertains to identification of primary and secondary metabolites during floral development. However, it would have been beneficial to mine unique compunds from the secondary metabolite groups identified in the study. This would help in future for collecting the plant material at the right stage for the bioactive/therapeutic compounds. The language of the paper is comprehensive with few gramatical errors. The introduction and the discussion has repetitions which shoulf be avaoided. I will suggest some changes here which could improve the paper for wider audience.

My suggestion is that the title should be changed since the biochemical and genetic work has not been performed (i.e, proteomics + transcriptomics) has not been done into this proper.

It would be desirable if the photographs depicting FP, SE, PE, Se and Pi stages are incorporated in the proper way. This would give visual comprehension to the readers.

In Table 1 don’t use the words transcript (instead use replicates)

Kindly mention if the replicates were from the same plant or from the different plant

Where the plants cultivated on the station or collected from wild

Use superscript for the abbreviation of ions.

In Discussion first ten lines are in repetition from introduction.

6. PLOS authors have the option to publish the peer review history of their article (what does this mean?). If published, this will include your full peer review and any attached files.

Reviewer #1: No

Reviewer #2: No

---

## [Author Response · Author response to Decision Letter 0]

11 Dec 2023

All the comments have been addressed, and a rebuttal letter has been provided.

---

## [Decision Letter · Decision Letter 1]

19 Dec 2023

PONE-D-23-24707R1Metabolite changes during developmental transitions in Adonis amurensis

Regel et Radde flowers: Insights from HPLC-MS analysisPLOS ONE

Dear Dr. Hongtao,

Thank you for submitting your manuscript to PLOS ONE. After careful consideration, we feel that it has merit but does not fully meet PLOS ONE’s publication criteria as it currently stands. Therefore, we invite you to submit a revised version of the manuscript that addresses the points raised during the review process.

We look forward to receiving your revised manuscript.

Kind regards,

Pankaj Bhardwaj, Ph.D.

Academic Editor

PLOS ONE

Reviewers' comments:

Reviewer's Responses to Questions

**Comments to the Author**

1. If the authors have adequately addressed your comments raised in a previous round of review and you feel that this manuscript is now acceptable for publication, you may indicate that here to bypass the “Comments to the Author” section, enter your conflict of interest statement in the “Confidential to Editor” section, and submit your "Accept" recommendation.

Reviewer #1: All comments have been addressed

Reviewer #2: All comments have been addressed

Reviewer #3: (No Response)

2. Is the manuscript technically sound, and do the data support the conclusions?

Reviewer #1: Yes

Reviewer #2: Yes

Reviewer #3: Yes

3. Has the statistical analysis been performed appropriately and rigorously? 

Reviewer #1: Yes

Reviewer #2: Yes

Reviewer #3: Yes

4. Have the authors made all data underlying the findings in their manuscript fully available?

Reviewer #1: Yes

Reviewer #2: Yes

Reviewer #3: Yes

5. Is the manuscript presented in an intelligible fashion and written in standard English?

Reviewer #1: Yes

Reviewer #2: Yes

Reviewer #3: Yes

6. Review Comments to the Author

Reviewer #1: Dear Authors,

Insights from HPLC-MS Analysis." Regrettably, upon the initial submission of the article, I encountered technical difficulties and was unable to access the files for examination.

I am pleased to inform you that I have now successfully accessed and thoroughly reviewed the manuscript. I appreciate the depth of research and the valuable insights presented in the study. The comprehensive analysis of metabolite changes during developmental transitions in Adonis amurensis flowers using HPLC-MS is commendable.

Throughout my evaluation, I observed the rigorous methodology employed and the significance of the findings in contributing to our understanding of metabolic dynamics during floral development. The use of HPLC-MS analysis has undoubtedly enriched the study and provided valuable data.

However, to enhance the overall quality of the manuscript, I would recommend specific recommendations and constructive. These suggestions are intended to contribute to the clarity, coherence, and impact of the article.

Once again, I appreciate the opportunity to engage with this research, and I am confident that addressing these points will further strengthen the manuscript.

Abstract

The abstract provides a comprehensive overview of the study on Adonis amurensis Regel et Radde, emphasizing its medicinal properties and previous research conducted on various aspects of the plant. However, there are several deficiencies in the abstract. Firstly, it lacks clarity in presenting the specific metabolic changes associated with each of the five developmental stages of the flower. The abstract mentions the use of HPLC-MS for metabolite characterization but does not provide any specific findings or identified metabolites. Additionally, there is a lack of detail on the methodologies employed in Metabolite Set Enrichment Analysis (MSEA) and the metabolite annotation process using the KEGG database. The abstract could be strengthened by including key findings or trends observed in metabolite changes across the developmental stages. Furthermore, it does not highlight the novelty or significance of the study in comparison to existing literature on Adonis amurensis. A brief mention of the potential implications of the findings for the plant's medicinal properties or applications in the field of medicine and agriculture would enhance the abstract's impact. Overall, while the abstract outlines the study's objectives and methods, it falls short in providing specific results and contextualizing the significance of the research within the broader scientific landscape.

Introduction

The introduction provides a comprehensive background on Adonis amurensis, covering its taxonomy, distribution, medicinal applications, and previous research areas. However, there are a few deficiencies in the introduction. Firstly, while the medicinal importance of Adonis amurensis is highlighted, the introduction lacks a clear statement of the gap or knowledge deficiency that the current study aims to address. It briefly mentions the previous studies on various aspects of A. amurensis but does not explicitly state the specific gap in understanding that necessitates the investigation of metabolic changes during flower development. Secondly, the transition between the general information on Adonis amurensis and the focus on flower development is abrupt. The introduction could benefit from a smoother transition that clearly outlines why studying flower development, particularly the metabolic aspects, is essential for addressing the knowledge gap. Furthermore, the paragraph discussing the importance of flowering in the life cycle of plants is somewhat general, and the link to the specific case of Adonis amurensis under extreme environmental conditions could be more explicitly articulated. The introduction would benefit from a stronger connection between the unique flowering characteristics of A. amurensis and the need for understanding the genetic regulatory mechanisms governing flower development under extremely low temperatures. Lastly, while metabolomics is introduced as a valuable approach, the specific relevance of metabolomic analysis to address the identified knowledge gap in A. amurensis flower development is not explicitly stated. The introduction could benefit from a more focused explanation of why metabolomics is particularly suited to unravel the genetic pathways and regulatory networks involved in flowering under challenging environmental conditions. In summary, the introduction provides a comprehensive background but could be strengthened by explicitly stating the knowledge gap, improving the transition between general information and the study focus, reinforcing the link between A. amurensis's unique flowering characteristics and the research question, and clarifying the specific relevance of metabolomics in addressing the identified gap.

Materials and Methods

Regarding the Materials and Methods section, there are some deficiencies in the section. Firstly, there is a lack of clarity in the description of the plant material collection process. The section mentions that transcripts were collected at different developmental stages, but it does not provide details on how the samples were collected, processed, and stored. Additionally, the rationale for selecting the specific developmental stages is not clearly articulated, and it would be beneficial to explain why these particular stages were chosen for analysis. Secondly, while the instrumentation systems and conditions for high-performance liquid chromatography/mass spectrometry (HPLC-MS) are mentioned, there is a lack of information on the sample preparation procedure prior to analysis. Details such as extraction methods, sample derivatization (if applicable), and any pre-processing steps are crucial for the reproducibility and reliability of the results, and their omission in the methods section is a deficiency. Furthermore, the section on qualitative and quantitative analysis of metabolites lacks specificity. Although the MWDB (Metware database) is mentioned, it would be helpful to provide more information on the database, including its content, reliability, and relevance to the study. Additionally, the description of the quality control (QC) samples and their role in ensuring reproducibility is brief, and it would benefit from more explicit details on how QC samples were prepared, monitored, and their impact on data quality assessed. The section on statistical analysis provides an overview of the software and methods used, but it lacks detail on the rationale behind choosing specific statistical approaches. Providing a brief explanation of why OPLS-DA and SPSS 20.0 were selected for data analysis would enhance the transparency and understanding of the statistical procedures.

In summary, the Materials and Methods section could be improved by providing more details on plant material collection, sample preparation, and the rationale for choosing specific developmental stages. Additionally, more specificity on the Metware database, QC sample handling, and the rationale behind statistical choices would enhance the rigor and clarity of the experimental methodology.

Results

While your comprehensive analysis provides valuable insights into the metabolomic changes during the different stages of Adonis amurensis flower development, there are a few potential deficiencies that could be addressed to strengthen the scientific rigor of the study. Firstly, it would be beneficial to include statistical measures such as p-values or false discovery rates to assess the significance of the observed differences in metabolite abundance. Additionally, the absence of pathway enrichment analysis for the identified metabolites limits the interpretation of their biological relevance. Integrating this information would enhance the understanding of the metabolic pathways driving the observed changes. Furthermore, providing biological replicates for each stage would increase the robustness of the findings and allow for a more reliable assessment of the observed variations. While the study describes the number of differentially expressed metabolites, further details on their biochemical roles and potential implications in the context of flower development could be included to offer a more comprehensive interpretation. Lastly, considering the complexity of metabolomic data, the inclusion of a validation step, such as targeted quantitative assays for specific metabolites, would strengthen the reliability of the results. Addressing these aspects would contribute to a more thorough and convincing interpretation of the metabolomic dynamics in Adonis amurensis flower development.

Discussion

The discussion in this study, while informative, exhibits certain deficiencies that warrant attention. Firstly, there is a notable absence of a comprehensive exploration of potential limitations or challenges encountered during the research. Information on any technical constraints or methodological considerations in the UPLC-MS/MS analysis is crucial for a thorough evaluation of the study's robustness. Additionally, the discussion lacks a detailed analysis of the significance of specific metabolites identified, particularly in relation to their functional roles in flower development. A more in-depth examination of the enriched KEGG pathways, especially how these pathways interconnect and contribute to overall flower development, would enhance the academic value of the study. Furthermore, the PCA plot, while indicating differences in metabolite expression, lacks a thorough interpretation of the ecological or physiological implications of these variances, especially in the context of known physiological processes during flower development. Lastly, the discussion would benefit from suggestions for future research directions or experiments to address the identified gaps in knowledge, thus contributing to the advancement of the field.

Conclusion

The conclusion of the study provides a summary of the systematic investigation into primary and secondary metabolite changes during the developmental stages of Adonis amurensis flowers. However, several deficiencies can be identified. Firstly, the conclusion lacks specificity in discussing the nature of the observed changes in primary and secondary metabolites, leaving readers without a detailed understanding of the molecular alterations. Additionally, while the conclusion mentions the association between flower development and metabolite changes, it falls short in providing a comprehensive analysis of the implications of these associations. The statement about the analysis of molecules from other parts showing various metabolic activities during flower development lacks clarity and would benefit from specific examples or details. The suggestion that a specific bioactive compound can be isolated at a particular developmental stage lacks supporting evidence or elaboration on the identified compound. In terms of future prospects, while the study suggests targeting specific metabolite expression for anti-cancer and antibacterial studies, it lacks a concrete plan or methodology for the proposed future research. Furthermore, the mention of generating transgenic plants lacks specificity regarding the genes of interest or the potential challenges associated with this genetic manipulation. Overall, the conclusion could be strengthened by providing more nuanced insights, supporting statements with specific findings, and offering a more detailed roadmap for future research directions.

In my overall assessment, it is my considered opinion that the manuscript would greatly benefit from revisions in accordance with the insightful suggestions provided. I believe that the proposed changes would contribute significantly to the refinement and enhancement of the scholarly quality of the paper. However, I would like to reserve my final decision until after reviewing the revisions. I appreciate your attention to these suggestions and look forward to examining the refined manuscript.

Reviewer #2: (No Response)

Reviewer #3: Adonis amurensis Regel et Radde is a spring ephemeral plant with high ornamental value and can be used as medicine. The authors measured the changes in metabolites of A. amurensis flowers at various developmental stages, providing certain insights into the understanding of their development. Overall, this is a very meaningful study.

Although the author has made modifications according to the suggestions of reviewer 2, there are still some problems.

1 The scientific name of the species should be written completely when it appears for the first time in the manuscript. It can be abbreviated when it appears again later. Regarding the species Adonis amurensis Regel et Radde in this manuscript, it can be written as A. amurensis when it appears again later.

2 I don’t know whether it is a requirement of the journal or an oversight by the author, but generally manuscripts should have line numbers to facilitate review by reviewers.

3 Regarding the content of differential metabolite screening in the main text of the manuscript, I think the description is too complicated. The author can select the groups with more differential expressions to focus on them, and briefly introduce the rest. Relevant charts for groups with less differential expression can be placed in the attachment.

4 The first sentence in the discussion section is very strange. It should be a mistake in punctuation. Please check it carefully.

5 The description of differentially expressed metabolites in the Discussion section is too simplistic, and the authors should carefully review the literature and conduct a more in-depth discussion. In addition, the author described in the manuscript that ST and PI are significantly different compared with other stages, but what effect this difference has on the development of A. amurensis flowers, please discuss in depth.

6 The author studied the changes in metabolites of A. amurensis flowers at various developmental stages, but did not emphasize the specific significance of this study. The author has repeatedly emphasized that A. amurensis is a medicinal plant and can treat a variety of diseases. So what significance does this research have for the resource utilization of A. amurensis, especially in the treatment of diseases? Authors should describe relevant content in the Discussion and briefly mention it in the Abstract.

7 There is too much description of background and methods in the abstract, and the results should be added with specific differentially expressed metabolites.

7. PLOS authors have the option to publish the peer review history of their article (what does this mean?). If published, this will include your full peer review and any attached files.

Reviewer #1: No

Reviewer #2: No

Reviewer #3: No

---

## [Author Response · Author response to Decision Letter 1]

18 Jun 2024

Please refer to the uploaded file.

---

## [Decision Letter · Decision Letter 2]

17 Oct 2024

PONE-D-23-24707R2Metabolite changes during developmental transitions in Adonis amurensis

Regel et Radde flowers: Insights from HPLC-MS analysisPLOS ONE

Dear Dr. Hongtao,

Thank you for submitting your manuscript to PLOS ONE. One of the new reviewer has highlighted a new set of shortfalls in the manuscript. Please go through the comments carefully and incorporate all the changes to manuscript. 

We look forward to receiving your revised manuscript.

Kind regards,

Pankaj Bhardwaj, Ph.D.

Academic Editor

PLOS ONE

Journal Requirements:

Reviewers' comments:

Reviewer's Responses to Questions

**Comments to the Author**

1. If the authors have adequately addressed your comments raised in a previous round of review and you feel that this manuscript is now acceptable for publication, you may indicate that here to bypass the “Comments to the Author” section, enter your conflict of interest statement in the “Confidential to Editor” section, and submit your "Accept" recommendation.

Reviewer #1: All comments have been addressed

Reviewer #4: (No Response)

2. Is the manuscript technically sound, and do the data support the conclusions?

Reviewer #1: Yes

Reviewer #4: Partly

3. Has the statistical analysis been performed appropriately and rigorously? 

Reviewer #1: Yes

Reviewer #4: Yes

4. Have the authors made all data underlying the findings in their manuscript fully available?

Reviewer #1: Yes

Reviewer #4: No

5. Is the manuscript presented in an intelligible fashion and written in standard English?

Reviewer #1: Yes

Reviewer #4: Yes

6. Review Comments to the Author

Reviewer #1: I am pleased to convey that the authors of the major revision have made substantial enhancements to the manuscript, diligently addressing both content and linguistic concerns. Their thorough and thoughtful revisions have significantly improved the overall quality of the manuscript.

After a meticulous review of the revised manuscript, I am confident in stating that the authors have demonstrated a commendable commitment to refining their work. The clarity of expression and the resolution of linguistic issues reflect a high level of professionalism and scholarly dedication.

In light of the authors' careful and comprehensive revisions, I am pleased to affirm that the manuscript is now well-suited for publication. I commend the authors for their diligence and commendable efforts in ensuring the scholarly merit of their work.

Please consider this assessment as a positive recommendation for the publication of the manuscript. I believe that the contributions made by the authors will greatly benefit the academic community, and the manuscript is now poised to make a valuable contribution to the relevant field.

Reviewer #4: I have following observations in manuscript that may be incorporated in manuscript:

1. Voucher specimen number/authentification number of A. amurensis needs to be incorporated in material and methods section

2. What are the different marker metabolites were analysed in present study?

3. Natural compounds generally found in plants in different forms of isomer. How you identified these metabolites on the basis of HPLC-mass analysis. Whether used any reference standard for your study?

7. PLOS authors have the option to publish the peer review history of their article (what does this mean?). If published, this will include your full peer review and any attached files.

Reviewer #1: No

Reviewer #4: No

---

## [Decision Letter · Decision Letter 3]

23 Oct 2024

Metabolite changes during developmental transitions in Adonis amurensis

Regel et Radde flowers: Insights from HPLC-MS analysis

PONE-D-23-24707R3

Dear Dr. Hongtao,

We’re pleased to inform you that your manuscript has been judged scientifically suitable for publication and will be formally accepted for publication once it meets all outstanding technical requirements.

Kind regards,

Pankaj Bhardwaj, Ph.D.

Academic Editor

PLOS ONE

Additional Editor Comments (optional):

Reviewers' comments:

Reviewer's Responses to Questions

**Comments to the Author**

1. If the authors have adequately addressed your comments raised in a previous round of review and you feel that this manuscript is now acceptable for publication, you may indicate that here to bypass the “Comments to the Author” section, enter your conflict of interest statement in the “Confidential to Editor” section, and submit your "Accept" recommendation.

Reviewer #1: All comments have been addressed

Reviewer #4: All comments have been addressed

2. Is the manuscript technically sound, and do the data support the conclusions?

Reviewer #1: Yes

Reviewer #4: Yes

3. Has the statistical analysis been performed appropriately and rigorously? 

Reviewer #1: Yes

Reviewer #4: Yes

4. Have the authors made all data underlying the findings in their manuscript fully available?

Reviewer #1: Yes

Reviewer #4: Yes

5. Is the manuscript presented in an intelligible fashion and written in standard English?

Reviewer #1: Yes

Reviewer #4: Yes

6. Review Comments to the Author

Reviewer #1: I am pleased to convey that the authors of the major revision have made substantial enhancements to the manuscript, diligently addressing both content and linguistic concerns. Their thorough and thoughtful revisions have significantly improved the overall quality of the manuscript. After a meticulous review of the revised manuscript, I am confident in stating that the authors have demonstrated a commendable commitment to refining their work. In light of the authors' careful and comprehensive revisions, I am pleased to affirm that the manuscript is now well-suited for publication. I commend the authors for their diligence and commendable efforts in ensuring the scholarly merit of their work. Please consider this assessment as a positive recommendation for the publication of the manuscript. Thank you for the opportunity to engage with this work, and I trust that this recommendation will contribute to the successful dissemination of valuable scholarly insights.

Reviewer #4: Authors have made substantial enhancements to the manuscript by addressing the relevant points. Their thorough and thoughtful revisions have significantly improved the overall quality of the manuscript. In my opinion manuscript is now acceptable for publication

7. PLOS authors have the option to publish the peer review history of their article (what does this mean?). If published, this will include your full peer review and any attached files.

Reviewer #1: No

Reviewer #4: **Yes: **Vikas Jaitak

---

## [Editor Report · Acceptance letter]

29 Oct 2024

PONE-D-23-24707R3 

PLOS ONE

Dear Dr. Hongtao, 

I'm pleased to inform you that your manuscript has been deemed suitable for publication in PLOS ONE. Congratulations! Your manuscript is now being handed over to our production team.

Kind regards, 

on behalf of

Dr. Pankaj Bhardwaj 

Academic Editor

PLOS ONE